# *Aeromonas* in South Asia: genomic insights into an environmental pathogen and reservoir of antimicrobial resistance

Nisha Singh [1,2,11], Rahma O. Golicha[1,3,11], Chetan Thakur[2], Mathew A. Beale [1], Matthew J. Dorman [1,4,5], Adrian Cazares [1], Alyce Taylor-Brown[6], Fatema-Tuz Johura[7], Mahamud-ur Rashid[7], Shirajum Monira[7], Fatema Zohura[7], Tahmina Parvin[7], Sazzadul Islam Bhuiyan[7], Marzia Sultana[7], Balvinder Mohan[2], Daryl Domman [8], Christine Marie George[9], Samuel Kariuki [3,12], Munirul Alam [7,12], Neelam Taneja[2,12] & Nicholas R. Thomson [1,10,12] ✉

Aeromonads are an ecologically versatile group of bacteria that cause infections in aquatic animals and are recognised as emerging human pathogens. Despite this, our understanding of *Aeromonas* diversity, especially the relationship between clinical and environmental strains, remains limited. Here, we present a genomic analysis of the *Aeromonas* genus, comprising 1853 genomes, and a detailed comparison of clinical and environmental strains from South Asia, including 996 newly sequenced genomes from Bangladesh and India. Phylogenetic analyses revealed that *Aeromonas* is a highly diverse genus, with no distinct clade separating clinical and environmental isolates. We identified 28 *Aeromonas* species and 905 novel sequence types, comprising 72.5% of the genomes. Notably, we show a high incidence of antimicrobial resistance (AMR) genes across all isolates, including against front and last-line antibiotics. Finally, we highlight frequent misidentification of *Aeromonas* as *Vibrio cholerae*, which is relevant to cholera-endemic regions where both genera co-exist and are associated with diarrhoeal disease. Our study underscores *Aeromonas* as an important environmental AMR reservoir and emerging multi-species pathogen capable of spilling over into human populations.

Aeromonads (within the *Aeromonadaceae* family) are Gram-negative, facultatively anaerobic bacteria with oxidase and catalase activity. *Aeromonas* spp. are abundant in aquatic habitats, with over 30 species identified[1]. At least 19 of these species are linked to human infections and are capable of causing significant diarrhoeal outbreaks. *Aeromonas* infections are common in children under 5 years old, especially in low- and middle-income countries (LMICs), where there is inadequate access to water and sanitation, resulting in high environmental

[1]Parasites & Microbes Programme, Wellcome Sanger Institute, Wellcome Genome Campus, Hinxton, Cambridge CB10 1SA, UK. [2]Department of Medical Microbiology, Post Graduate Institute of Medical Education & Research (PGIMER), Chandigarh 160012, India. [3]Kenya Medical Research Institute (KEMRI), Nairobi, Kenya. [4]School of Mathematical and Statistical Sciences, College of Science and Engineering, University of Galway, University Road, Galway H91 TK33, Ireland. [5]Department of Microbiology, Moyne Institute of Preventive Medicine, School of Genetics and Microbiology, Trinity College Dublin, Dublin 2 D02 PN40, Ireland. [6]Children's Health Queensland Hospital and Health Service, South Brisbane, QLD, Australia. [7]Infectious Diseases Division, icddr,b (International Centre for Diarrhoeal Disease Research, Bangladesh), Dhaka, Bangladesh. [8]Bioscience Division, Los Alamos National Laboratory, Los Alamos, NM, USA. [9]Department of International Health, Johns Hopkins Bloomberg School of Public Health, Baltimore, MD, USA. [10]London School of Hygiene and Tropical Medicine, London WC1E 7HT, UK. [11]These authors contributed equally: Nisha Singh, Rahma O. Golicha. [12]These authors jointly supervised this work: Samuel Kariuki, Munirul Alam, Neelam Taneja, Nicholas R. Thomson. ✉e-mail: nrt@sanger.ac.uk

exposure. The most common symptoms of *Aeromonas* infection include diarrhoea (which can be mistaken for cholera[2,3]), localised soft tissue infection, and bacteraemia[4–8]. Bacteraemia is most common in patients who have underlying health conditions, such as hepatobiliary disease, cancer, or diabetes[9]. There appear to be geographic differences in prevalence, with <1% of cases of moderate to severe diarrhoea (MSD), attributable to *Aeromonas* in four African sites and Kolkata, India, whereas *Aeromonas* was implicated in >22% of MSD cases in children from Karachi, Pakistan, and Mirzapur in Bangladesh[10]. *Aeromonas* spp. comprise important pathogens of fish and aquatic animals, causing diseases such as haemorrhagic septicaemia, furunculosis, and motile *Aeromonas* septicaemia (MAS) in species like salmonids, catfish, and tilapia, often leading to high mortality and significant economic consequences for aquaculture[11–17]. Additionally, *Aeromonas* spp. have also been isolated from terrestrial animals and are recognised as zoonotic, with human infections commonly occurring through contact with infected animals or contaminated water, especially via wounds[18–20].

Pathogenicity in Aeromonads is complex; they possess a variety of virulence factors that contribute to biofilm formation, cell adherence, invasion, and cytotoxicity. These include polar and lateral flagella[21,22], adhesins[23], lipopolysaccharides, iron-binding systems[24,25], and numerous extracellular toxins and enzymes[26] secreted by various systems such as type 2 and type 3 secretion systems[27,28]. Additionally, quorum-sensing systems play a crucial role in colonisation and disease development[29–31].

Notwithstanding the above, *Aeromonas* spp. are frequently misidentified as *Vibrio cholerae* in clinical and environmental samples. This is explained by sharing many biochemical and microbiological properties with other genera such as *Aerobacter*, *Pseudomonas*, *Escherichia*, *Proteus*, and *Vibrio*[3,32]. Both *Vibrio* and *Aeromonas* genera are oxidase-positive, able to thrive in aquatic and marine systems and exhibit similar colony morphologies on Thiosulphate Citrate Bile Salt Sucrose (TCBS) medium routinely used to select for *V. cholerae*[33]. These misidentifications pose serious epidemiological challenges, especially in regions where *V. cholerae* is endemic. Misidentification can not only lead to misreporting of cholera cases, but also a failure to implement necessary public health control measures. Hence, it is important to consider *Aeromonas* spp. in cholera surveillance and control efforts in endemic regions.

Accurate identification of *Aeromonas* spp. remains challenging due to overlapping phenotypic traits and the limited specificity of conventional biochemical tests and selective media, which often produce inconclusive results[34–36]. Molecular methods such as 16S rRNA sequencing are helpful but complicated by intragenomic variability, reducing species-level accuracy[37,38]. Emerging technologies like MALDI-TOF MS offer rapid and more reliable genus- and species-level identification[37,39,40], while whole-genome sequencing provides definitive species-level resolution and accurate subtyping. Currently, a tiered diagnostic approach combining initial MALDI-TOF screening with phenotypic assays and, when necessary, molecular confirmation at reference laboratories is recommended to ensure accurate detection and effective epidemiological surveillance of *Aeromonas* spp.[40]. It is likely that, as for many other pathogens, genomic approaches will replace these techniques where accessible and in settings with well-developed supply chains.

Aeromonads are known to exhibit high levels of antimicrobial resistance (AMR), especially against β-lactam antibiotics. This resistance is primarily attributed to chromosomally encoded inducible β-lactamases in species such as *A. hydrophila*, *A. sobria*, and *A. salmonicida*, conferring resistance to antibiotics including ampicillin, bacitracin, cefoxitin, and piperacillin/tazobactam[41–45]. Recent genome data from 447 *Aeromonas* isolates from Pakistan revealed that human-associated *Aeromonas* in both clinical cases and age-matched controls were highly diverse and displayed a high level of AMR, particularly

extended-spectrum beta-lactamases (ESBLs)[46]. Moreover, *Aeromonas* serve as natural reservoirs for mobile colistin resistance genes (*mcr*)[47,48], raising concerns about their role in spreading resistance to last-resort antibiotics like colistin across bacterial populations in aquatic and clinical environments. Looking into determinants that may explain the difference in clinical presentations, this study looked across an array of known or putative virulence factors, of which only two, *maf2* and *lafT*, were weakly associated with the MSD cases. However, by focusing on isolates from humans, this study was unable to establish whether human pathogenic strains were a specialised subset of those found in the environment or if *Aeromonas* spp. infections are truly opportunistic and linked to environmental exposure[46].

Given the ubiquitous nature of Aeromonads in the environment, their potentially emerging role in MSD in South Asia, and the fact that they are often cultured from samples taken from patients with suspected cholera, we created a baseline population genomic snapshot of *Aeromonas* species from clinical and environmental samples. We sequenced two collections of *Aeromonas* isolates from *V. cholerae* endemic regions of Bangladesh (2004–2016), and Northern India (2020–2023). These isolates were obtained from environmental sources—including drinking water, ponds, rivers, lakes and drains, as well as from human faeces, together comprising 996 novel *Aeromonas* genomes. We compared their genetic diversity, predicted virulence gene profiles, and AMR profiles, and contextualised this with genomes from a global *Aeromonas* population dataset.

## Results

### Species diversity and distribution of *Aeromonas* spp. genomes

To examine the genomic diversity of *Aeromonas* strains across clinical and environmental settings in South Asia, we included the largest and only available clinical genome datasets from Pakistan and environmental datasets from India and Bangladesh (this study). To place our study into a global context, we also incorporated genomes representing broad geographic and species diversity within the *Aeromonas* genus. In total, we assembled 1853 *Aeromonas* genomes, including 996 newly sequenced genomes, of which 198 (10.68%) were mistakenly collected as *Vibrio cholerae*—133 from suspected cholera cases/households contacts in the ChoBI7 programme in Dhaka city[49] and 65 from an environmental surveillance study for *V. cholerae* conducted in Bangladesh. Further, we generated 798 *Aeromonas* genomes from isolates taken from 274 water samples collected in Northern India. We contextualised our dataset using public archives, including 441 genomes of clinical origin from children with moderate to severe diarrhoea and corresponding healthy controls, taken in Pakistan[46], as well as 416 globally distributed[50] *Aeromonas* genomes. The latter were derived from a wide range of sample types, including human stool, urine, wounds, as well as aquatic environments (e.g., rivers and wastewater), and animals (e.g., fish, duck, dog). The largest proportion of publicly available genomes originated from Denmark ($n = 100$), followed by the United States ($n = 71$) and China ($n = 62$) (see Supplementary Table 1 for detailed information).

We used GTDB-Tk (Genome Taxonomy Database) to perform species assignment, identifying 28 *Aeromonas* spp. across the 1853 genomes, of which eight species dominated. *A. caviae* ($n = 608$) was the most numerous in our collection, followed by *A. veronii* ($n = 566$), *A. dhakensis* ($n = 186$), *A. salmonicida* ($n = 176$), *A. hydrophila* ($n = 124$), *A. enteropelogenes* ($n = 105$), *A. jandaei* ($n = 25$), *A. sanarellii* ($n = 16$), with the remaining 47 *Aeromonas* genomes representing 20 other *Aeromonas* species (Fig. 1; summarised in Supplementary Fig. 1a). To understand how our species predictions mapped to established species we used Average Nucleotide Identity (ANI) to evaluate their genetic relatedness. The intra-species ANI values ranged from 94.06% to 100%, falling slightly below the established definition for a single species (>95% ANI), in some instances. Conversely, inter-species ANI values ranged from 79.66% to 95.51% with higher-than-expected ANI

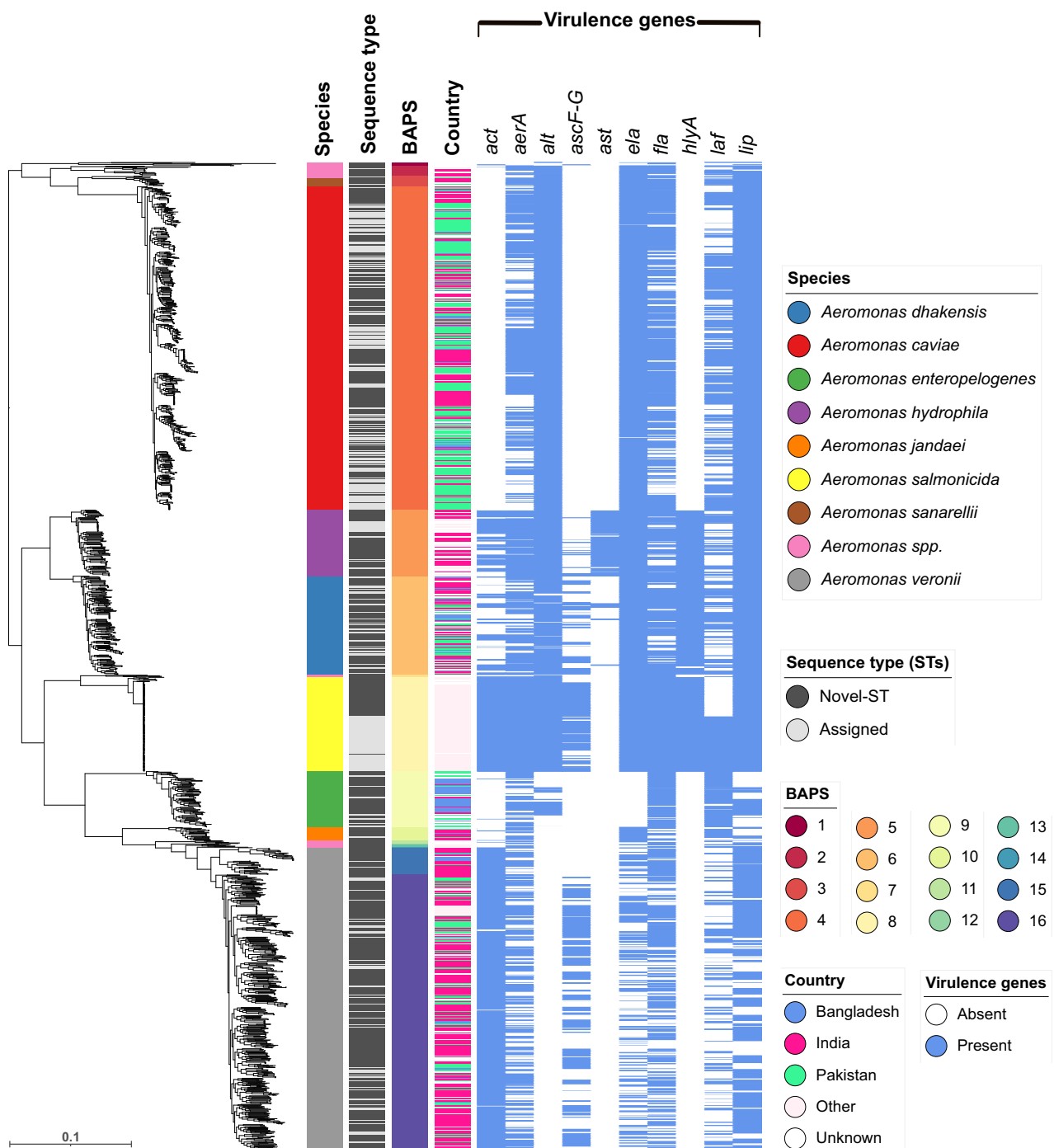

**Fig. 1 | Taxonomic, genetic and geographic diversity of 1853 *Aeromonas* species genomes.** A maximum likelihood phylogenetic tree based on 1969 core genes for *Aeromonas* spp. genomes. Refer to the key for the interpretation of the colour strips alongside. The scale indicates an evolutionary distance of 0.1 nucleotide substitutions per site. BAPS refers to Bayesian Analysis of Population Structure, used here to delineate genetic clusters.

and lower levels of expected divergence between *A. bestiarum* and *A. piscicola* (95.14–95.51%) (Supplementary Fig. 2).

Looking at the geographic distribution of *Aeromonas* species across both clinical and environmental samples from India, Pakistan and Bangladesh, where we had the densest sampling (1438 *Aeromonas* genomes; Supplementary Fig. 1b), *Aeromonas* spp. found across all three South Asian countries included *A. caviae, A. dhakensis, A. enteropelogenes, A. jandaei, A. veronii* and *A. sanarellii*, with *A. sanarellii* and *A. jandaei* being rare in all three countries. Whilst overall there were similarities in the relative abundance of the different species, despite

differences in sampling depth and strategy (see methods), there were also clear differences. In particular, *A. caviae* predominated in Bangladesh and Pakistan (31.81%, 63/198 and 64.39%, 284/441; respectively), whereas *A. veronii* dominated the samples collected in India (48.31%, 386/799 of isolates). Other species, such as *A. schubertii*, were exclusively seen in Bangladesh (*n* = 1), while *A. media* (*n* = 4) and *A. rivipollensis* (*n* = 5) were exclusively seen in India (Supplementary Fig. 1b). What is more, 129 environmental samples from Northern India included between 1–5 different *Aeromonas* species (Supplementary Fig. 3a). For countries where we had less dense sampling, *A. hydrophila*

was the most common in the United States (42.25%, 30/71), whereas *A. salmonicida* was the dominant species in the samples from Denmark (99%, 99/100) and China (96.77%, 60/62), (Supplementary Table 1).

In India, all samples were collected as part of a cholera surveillance study. Samples were taken from known cholera hotspots and at times of the year when cholera cases were more common. All water samples were cultured on *V. cholerae* selective media (TCBS; see methods). Using MALDI-TOF, it was possible to look at the *Aeromonas* spp. positivity rate, as well as for the presence of a range of other enteric pathogens (Supplementary Fig. 4). Out of 408 collected water samples from Northern India, 274 samples (67.15%) tested positive for *Aeromonas* spp., 145 (35.53%) for *V. cholerae*, 112 (27.45%) for *E. coli*, 198 (48.52%) for *Klebsiella* spp. and 35 (8.57%) for non-cholera *Vibrio* spp. (8.57%, 35/408; including species *V. fluvialis*, *V. navarrensis*, *V. cidicii*, *V. injensis*, and *V. metschnikovii*).

Of the samples positive for *Aeromonas* spp., *V. cholerae* was also isolated from 91 samples (33.57%, 91/274). These co-occurrences were primarily found in river samples (75/91), followed by ponds (13/91) and canals (4/91), across different regions of Northern India. A total of seven *Aeromonas* species: *A. veronii*, *A. caviae*, *A. hydrophila*, *A. dhakensis*, *A. jandaei*, *A. enteropelogenes*, and *A. sanarellii*−were identified as cohabiting with *V. cholerae*. However, there was no clear association between the co-occurrence of *V. cholerae* with any one specific *Aeromonas* species; these data largely mirrored the relative isolation rates for *Aeromonas* species from Northern India (Supplementary Fig. 3b; Supplementary Fig. 3c).

## Genomic taxonomy of *Aeromonas* spp

To infer the phylogeny of the genus, we standardised the annotation of our underlying dataset using PROKKA and then defined the *Aeromonas* spp. pangenome using Panaroo. This comprised 47,542 coding sequences (CDS), of which 1969 were defined as core (found in more than 99% of genomes) and 118 as soft core (found in 95−99% of genomes), with 45,455 accessory genes (found in 0−95% of genomes). We constructed a maximum likelihood phylogenetic tree from a core gene alignment (Fig. 1; see methods). We used Bayesian Analysis of Population Structure (BAPS, implemented in fastBAPS) to delineate genomic clusters and correlated these with ANI values and the species assignments from GTDB-Tk. From BAPS, whilst most taxa fell on deeply branching nodes in the *Aeromonas* spp. tree with high concordance with the ANI-defined species blocks (Fig. 1; Supplementary Fig. 2), *A. veronii* fell across two BAPS clusters (15 and 16). Conversely, BAPS clusters 1, 2, 3, and 7 each span multiple species of *Aeromonas* (Supplementary Table 2). These inconsistencies may reflect differences between the way the BAPS model clusters genomes, which, unlike ANI, is not distance-based.

Next, to determine how much of the diversity within known taxonomic groups had been previously observed, we constructed in silico MLST[51] profiles. Of the 1853 MLST profiles generated, only 495 (26.71%) genomes were assigned to 224 known MLST allelic profiles, with 1343 genomes representing 905 novel sequence types (STs). After assignment by pubmlst.org (https://pubmlst.org/organisms/aeromonas-spp), 1838 of our *Aeromonas* spp. genomes were assigned to 1129 STs (15 genomes remained unassigned; see Supplementary Data 1), now accessible through the *Aeromonas* MLST database[52]. Of note, these novel STs define discrete branches within all species, while 836 STs represented singletons, with clusters of genomes belonging to the same ST being rare. The exceptions to this were ST-2 (*n* = 103) and the newly defined ST-1799 (*n* = 60) for *A. salmonicida*, as well as ST-2398 (*n* = 20) and ST-251 (*n* = 19), representing *A. caviae* and *A. hydrophila* genomes, respectively.

## Virulence gene distribution in clinically and environmentally derived *Aeromonas* spp

Several virulence genes have been previously described amongst *Aeromonas* species, including the haemolytic toxins: aerolysin-related cytotoxic enterotoxin (*act*)[41,53,54], heat-labile cytotonic enterotoxin (*alt*)[53–55], heat-stable cytotonic toxins (*ast*)[53,55], hemolysin (*hlyA*)[56], cytotoxin aerolysin (*aerA*)[53,55,56] as well as polar flagellum (*fla*)[55,57,58], lateral flagella (*laf*)[59,60], the zinc-and-iron-dependent metallo-endopeptidase elastase (*ela*)[57], the lipase (*lip*)[41,55] and the type III secretion system (TTSS) encoded by *ascF-G*[57]. To determine whether any of the known virulence genes are linked to the particular *Aeromonas* species, we used in silico PCR to understand the distribution of these genes across 1853 *Aeromonas* species genomes.

We analysed patterns of virulence gene distribution using principal component analysis (PCA) and hierarchical clustering (Fig. 2a; Fig. 2b). The PCA revealed that certain species, such as *A. hydrophila*, *A. dhakensis*, *A. salmonicida*, and *A. veronii*, formed relatively well-defined clusters, suggesting distinct virulence profiles (Fig. 1; Fig. 2a). Some overlap was observed between the phylogenetically related *A. hydrophila* and *A. dhakensis*, as well as between the phylogenetically distant *A. hydrophila* and *A. salmonicida*. Genes such as *aerA*, *fla*, *laf*, and *lip* were found across nearly all species. For instance, *aerA* was present in over 60% of *A. caviae*, *A. dhakensis*, and *A. veronii*, and in over 95% of *A. hydrophila* and *A. salmonicida* genomes. Similarly, *fla* was widely distributed, occurring in 70−90% of genomes across most species. Consistent with previous studies[61,62], *hlyA* and *ast* were more species-specific, with *hlyA* almost exclusively found in *A. dhakensis* (99.5%) and *A. hydrophila* (100%), and *ast* present in 95.2% of *A. hydrophila* genomes but nearly absent elsewhere. *A. caviae* (*n* = 608) and *A. salmonicida* (*n* = 176) carried the highest proportions of key virulence genes; all *A. caviae* genomes encoded *alt* and *lip*, with nearly all (99.7%) possessing *ela*, while *A. salmonicida* uniformly carried *aerA*, *alt*, *ela*, *fla*, *lip*, and *act*. Genes *ela* and *lip* were also prevalent in *A. dhakensis* (100%), *A. hydrophila* (100%) and *A. sanarellii* (93.8%), while *A. enteropelogenes* and *A. jandaei* lacked several virulence genes (Fig. 1; summarised in Fig. 3a).

The Type III secretion system genes (*ascF-G*) were found more often in *A. salmonicida* (80%, 141/176), *A. veronii* (51%, 288/566), *A. dhakensis* (42%, 78/186), and *A. hydrophila* (17%, 21/124). Gene *hlyA* was present in all isolates of *A. salmonicida*, *A. dhakensis*, and *A. hydrophila*, but absent in *A. jandaei*, *A. enteropelogenes*, *A. sanarellii*, *A. caviae*, and *A. veronii*, representing a potential molecular marker for differentiating between species (Fig. 2b). Similarly, the enterotoxin-encoding *ast* gene was found only in *A. hydrophila* (95%, 118/124) and at low levels in *A. dhakensis* (6%, 11/186), highlighting its potential for species-specific identification. We noted, a high prevalence of the combination of the cytotoxic enterotoxin (*act*) and cytotonic heat-labile enterotoxin (*alt*) in *A. salmonicida* (*alt* = 100%, 176/176; *act* = 99%, 175/176), *A. hydrophila* (*alt* = 100%, 124/124; *act* = 72%, 90/124), and *A. dhakensis* (*alt* = 94%, 176/186; *act* = 29%, 55/186). We also observed variation in the distribution of *laf*, suggesting that this gene had been gained or lost multiple times across the phylogeny (Fig. 1); for example, it was present in *A. salmonicida* ST-2 but absent from ST-1799.

Lastly, we investigated whether virulence gene presence varied within species depending on sample origin. For this, we focused on South Asia (Bangladesh, India and Pakistan), since here we had 1438 *Aeromonas* spp. genomes confirmed to originate from clinical or environmental samples. As shown in Supplementary Fig. 5a and 5b, no obvious association was observed between species, sample origin, and virulence gene presence. This was also true when looking at the phylogeny of *A. caviae* taken from South Asia (*n* = 577), where we had almost equal representation from clinical (*n* = 286; of which 153 were from individuals with moderate to severe diarrhoea, while 133 were asymptomatic) and environmental (*n* = 291) *Aeromonas* genomes (Supplementary Fig. 6). Of the 11 BAPS clusters, 10 contained both clinical and environmental *Aeromonas* genomes; the only exception was cluster-4.10, which included just five isolates in total, all from environmental sources. Sublineages containing a mixture of both clinical and environmental isolates were also identified in the other

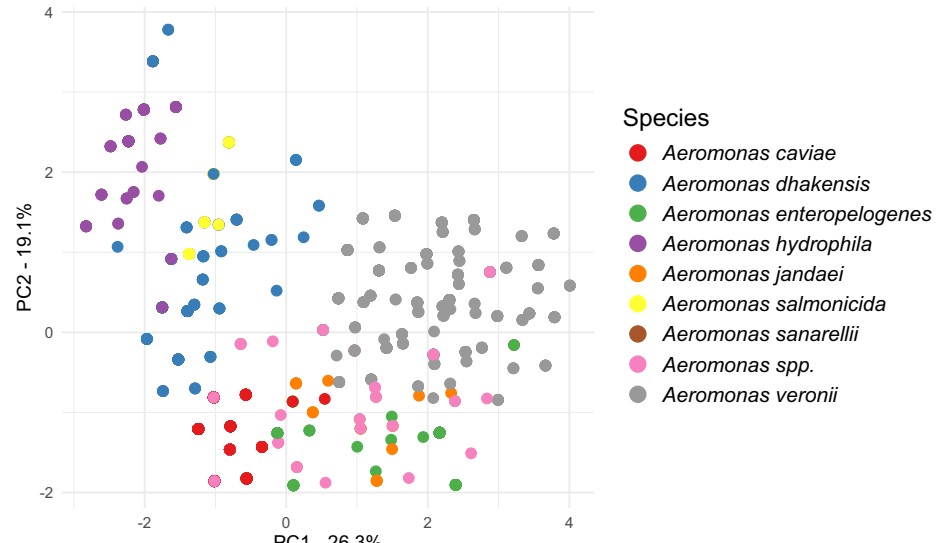

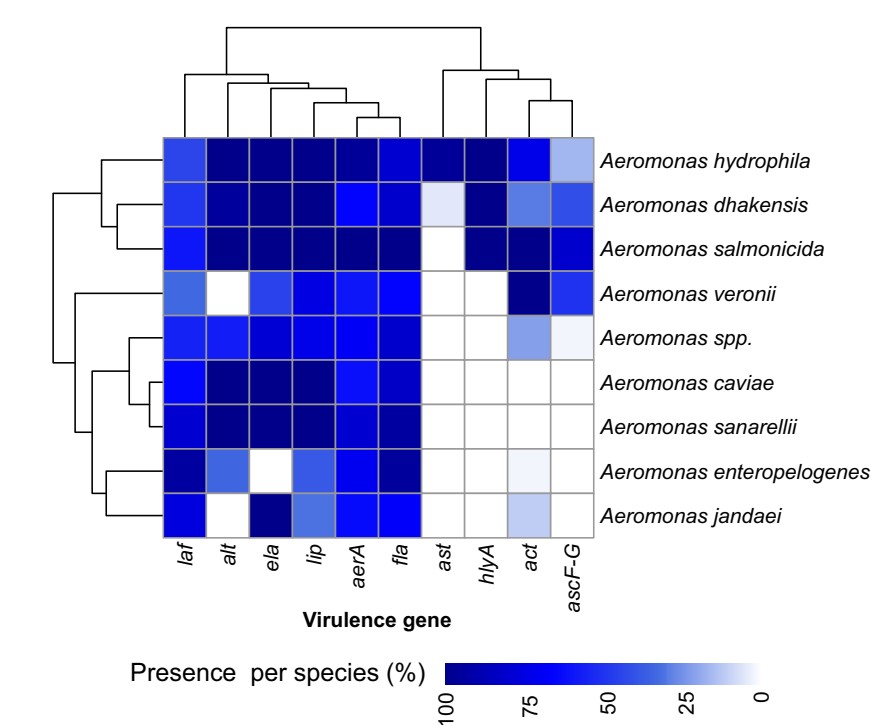

**Fig. 2 | Distribution of virulence genes across 1853 *Aeromonas* species genomes.**
**a** Principal component analysis (PCA) of virulence genes across different *Aeromonas* spp. (see key for coloured dots representing species). **b** Hierarchical clustering of virulence genes across *Aeromonas* spp., with colours representing the percentage presence of genes (see key for scale).

abundant *Aeromonas* species (*A. veronii, A. dhakensis, A. enteropelogenes*, and *A. hydrophila*; Supplementary Table 3).

### In silico prediction of AMR

To understand if there were differences in AMR profiles across or within species, we inferred the genotypic AMR profiles for all the genomes included in this study. Overall, we found 162 discrete AMR genes, conferring resistance to 16 different antimicrobial classes, including β-lactams, sulphonamides, tetracycline, and aminoglycosides (Fig. 3b, summarised in Supplementary Table 4). Of the 162 AMR

genes identified, those conferring resistance to β-lactam antibiotics were the most common, representing 40.74% (66/162) of the genes. Within this group, Class D oxacillinases were the most common, present in 94.5% of genomes, with genes such as $bla_{OXA-427}$ (44.6%, 826/1853) and $bla_{OXA-12}$ (32.7%, 605/1853) being particularly abundant. This was followed by Class C cephalosporinases, which were frequently found in 71.3% of genomes, including $bla_{MOX-6}$ (18.9%, 350/1853) and $bla_{FOX-2}$ (9.6%, 177/1853). Class B metallo-β-lactamases were detected in 55.4% of genomes, dominated by $bla_{CEPH-A3}$ (22.1%, 410/1853) and $bla_{cphA4}$ (9.6%, 177/1853). In contrast, Class A β-lactamases were less

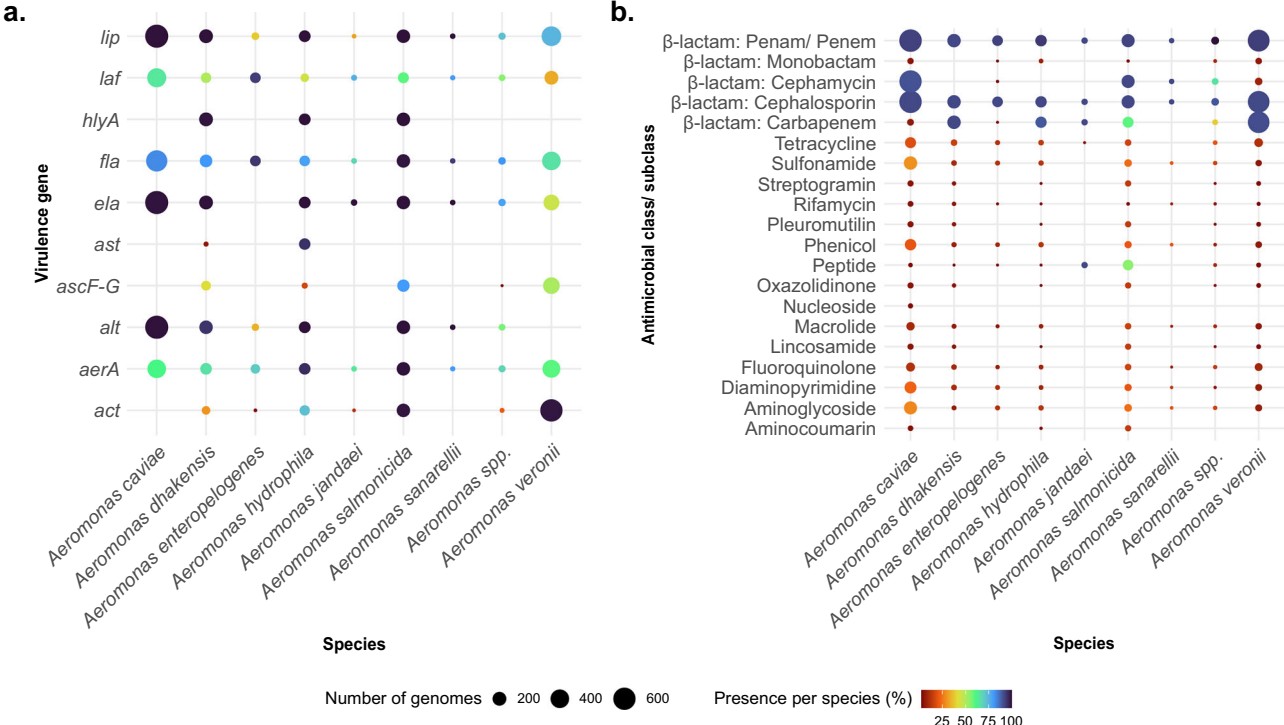

**Fig. 3 | Distribution of virulence genes and drug classes corresponding to antimicrobial resistance across 1853 *Aeromonas* species genomes.** The *x*-axis represents the most abundant *Aeromonas* species, with less abundant species grouped as *Aeromonas* spp., and the *y*-axis denotes virulence genes in (**a**) and antimicrobial classes corresponding to AMR genes in (**b**). Each dot is coloured based on the gene abundance per species (see scale), and the size of the dots corresponds to the number of genomes harbouring the gene.

common, identified in 5.07% of genomes, with genes such as $bla_{KPC-1}$ (0.8%, 15/1853) and $bla_{RSA-1}$ (1.1%, 20/1853) among the more notable types (Supplementary Table 5). Of note, mobile colistin resistance genes (*mcr*) were detected in several species. These included *A. jandaei* (100%, 25/25), all of which carried *mcr-7.1*; *A. piscicola* (100%, 1/1) harbouring *mcr-3* and *mcr-3.8*; *A. salmonicida* (59.1%, 104/176) carrying both *mcr-3* and *mcr-3.12*; *A. media* (28.6%, 2/7) with *mcr-3* and *mcr-3.6*; *A. sobria* (16.7%, 1/6) harbouring *mcr-3* and *mcr-3.12*; *A. veronii* (1.94%, 11/566) with *mcr-3*, *mcr-3.12*, *mcr-3.3*, *mcr-3.6*, and *mcr-3.8*; *A. caviae* (1.32%, 8/608) carrying *mcr-3*, *mcr-3.12*, *mcr-3.3*, and *mcr-3.6*; and *A. hydrophila* (0.81%, 1/124), which harboured *mcr-5* (Supplementary Table 4; see Supplementary Data 1). Only three genomes in our collection lacked any known AMR gene.

Looking more closely at the subset of 1438 South Asian genomes, we found that genotypic AMR patterns varied significantly across India, Bangladesh, and Pakistan (Fig. 4a). *Aeromonas* isolates from Pakistan exhibited the highest prevalence of resistance to β-lactams (cephamycin: 65.8%, 290/441), sulphonamides (24.3%, 107/441), diaminopyrimidines (22.4%, 99/441), aminoglycosides (22.2%, 98/441), tetracyclines (18.4%, 81/441), phenicols (16.3%, 72/441), and fluoroquinolones (12.9%, 57/441). Indian isolates showed comparatively lower prevalence of resistance gene presence amongst the three South Asian countries, with the highest resistance observed for carbapenems (69.2%, 553/799); macrolides (7.8%, 62/799); rifamycins (3.3%, 26/799); and oxazolidinones, pleuromutilins, and streptogramins (4.4%, 35/799). *Aeromonas* from Bangladesh had the lowest prevalence of resistance, with no genes conferring resistance to lincosamides, oxazolidinones, pleuromutilins, streptogramins, and nucleosides. Notably, all genomes across the three countries were predicted to carry cephalosporin resistance genes. Among these, 4.5% of the isolates (65/1438; 14 clinical and 51 environmental) carried genes encoding third-generation extended-spectrum β-lactamases (ESBLs), such as $bla_{CTX-M}$, $bla_{GES}$, $bla_{PER}$, $bla_{TEM}$, and $bla_{VEB}$.

To further refine our understanding beyond drug class-level resistance, we assessed the genetic differences in AMR gene profile between clinical and environmental isolates among the 1438 genomes using PCA and NMDS. This revealed considerable overlap between both groups within each species (Fig. 4c and Supplementary Fig. 7a). Notably, species like *A. caviae* and *A. veronii* formed distinct clusters, likely due to differences in AMR gene diversity (Supplementary Fig. 7b and Supplementary Table 6). Environmental isolates generally exhibited a greater diversity in AMR genes carriage than the clinical isolates across *Aeromonas* species (Supplementary Table 7), except in *A. caviae*, where clinical isolates contained more unique determinants (18 vs. 7) but also had largest overlap (48 shared genes; Supplementary Fig. 8). Notably, *A. veronii* and *A. dhakensis* environmental isolates carried extensive AMR repertoires (45 and 22 genes, respectively) with several shared with the clinical strains, suggesting potential environmental reservoirs (Supplementary Table 7). Shared genes across species commonly included *tet*(A), *tet*(E), $bla_{OXA}$ and $bla_{MOX}$ variants, *sul1/sul2*, and *qnr* variants, indicating frequent exchange between strains occurring in the aquatic and clinical settings, while certain ESBLs and carbapenemases (e.g., $bla_{CTX-M-15}$, $bla_{VIM-6}$) remained confined to the clinical isolates included in this study. Furthermore, certain AMR genes exhibited notable differences in prevalence between the two groups (Fig. 4d); clinical isolates had a higher proportion of $bla_{OXA-427}$ (64.7%, 288/445), $bla_{MOX-6}$ (35.7%, 159/445), and *sul1* (17.3%, 77/445), compared to environmentally derived isolate genomes. In contrast, environmental isolates showed higher proportions of $bla_{OXA-12}$ (43.9%, 435/990), $bla_{OXA-724}$ (18.3%, 181/990), $bla_{CEPHA3}$ (28.7%, 284/990) and $bla_{cphA4}$ (12.7%, 126/990) (Fig. 4d). Of note, one Indian environmentally derived *A. sanarellii* isolate was predicted to be multidrug resistant by possessing genes to seven drug classes including: aminoglycoside, β-lactam, diaminopyrimidine, macrolide, phenicol, rifamycin, and sulphonamide (see Supplementary Data 1).

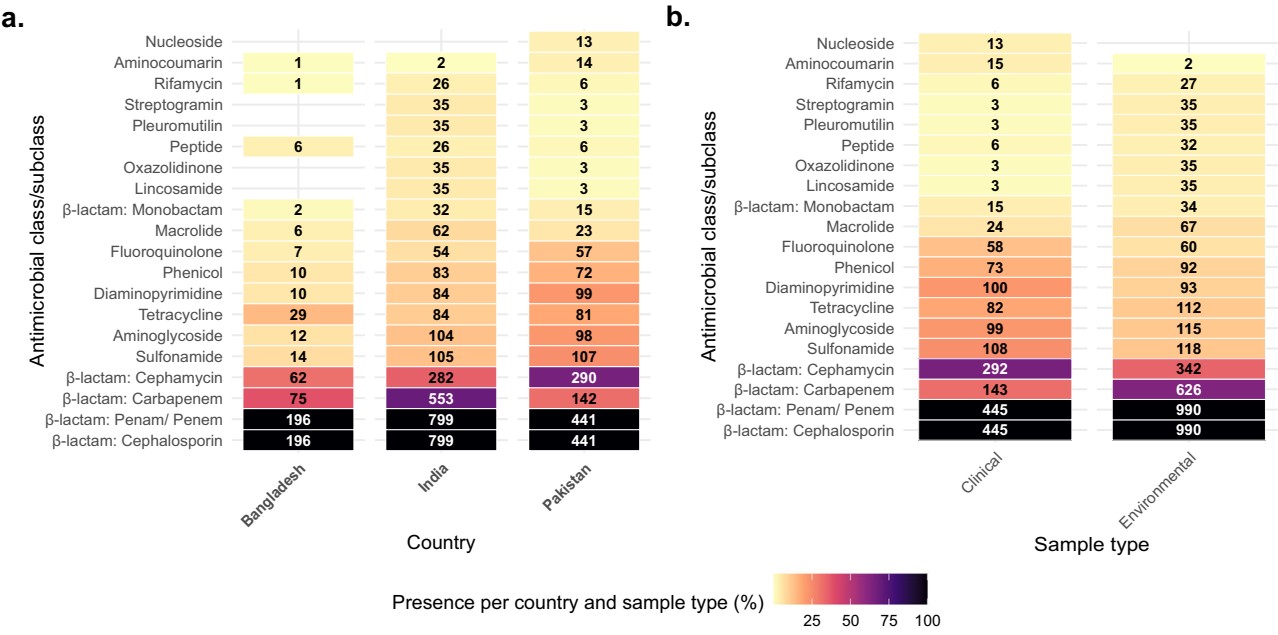

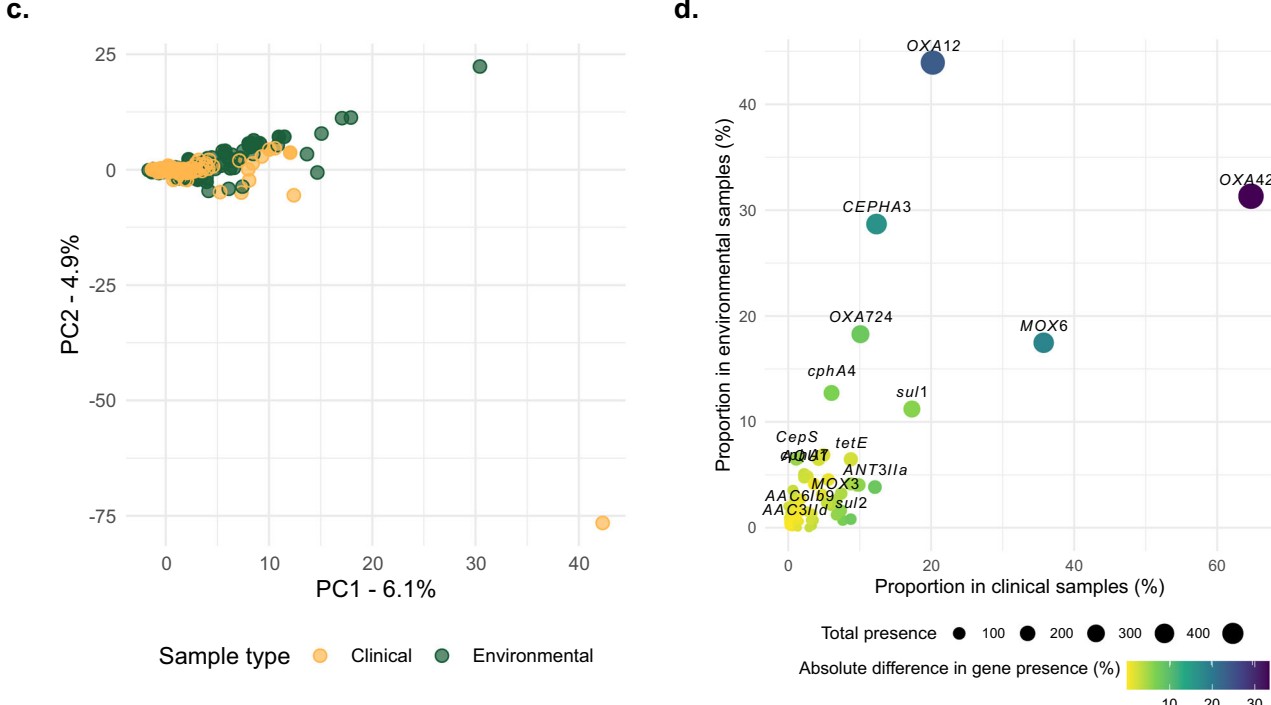

**Fig. 4 | Distribution of drug classes corresponding to antimicrobial resistance genes across 1436 *Aeromonas* species genomes in South Asia, including Bangladesh, India, and Pakistan.** The *x*-axis represents the country of origin in (**a**) and the sample type (clinical vs. environmental) in (**b**), while the *y*-axis denotes antimicrobial classes corresponding to resistance genes. The heatmap is colour-coded to indicate the percentage presence of AMR genes per country and sample type, with numeric indicators above each tile denoting the number of genomes carrying the gene within that category. **c** Principal component analysis (PCA) illustrates the clustering of genes in clinical and environmental isolates (see key for coloured dots). **d** Absolute difference in gene presence between clinical and environmental isolates, with yellow-green indicating minimal or no difference and blue-violet representing genes with significant variation in presence across sample types.

## Discussion

We initiated this study to provide a genomic framework for understanding the relationship between *Aeromonas* spp. from the natural aquatic environment and those isolated from clinical samples linked to disease. We also wanted to understand the nature and diversity of *Aeromonas* spp. co-inhabiting the same natural environment with *Vibrio cholerae*, in endemic settings. We included a range of isolates and genomes generated here, as well as published Aeromonads

genomes from moderate to severe diarrhoea cases and matched controls from Pakistan[46]. We investigated the genetic diversity, prevalence of virulence and antibiotic resistance genes amongst Aeromonads genomes from households and freshwater in multiple countries. In doing so, we have provided the most comprehensive view of this genus thus far.

Among 1853 genomes included in this study, we identified 28 *Aeromonas* species, including *A. veronii*, *A. caviae*, *A. dhakensis*, *A.

*enteropelogenes*, *A. salmonicida*, and *A. hydrophila*, several of which are known human or fish pathogens[13,16,17,20,34,63,64]. *A. dhakensis* and *A. enteropelogenes* were prevalent in Bangladesh, comprising approximately 22% and 29% of isolates, respectively, but were less common in the neighbouring countries. Other notable geographic signals include the psychrophilic *A. salmonicida*, seen largely in cooler countries such as Denmark (56%) and certain regions of China (35%, primarily from salmon facilities), where most commercial fish farming is done[65]. Here, the majority of the *A. salmonicida* were collected from infections of trout, *Oncorhynchus mykiss*, known to be highly susceptible to *A. salmonicida* infections. Similarly, 20% of *A. hydrophila* described here originated from diverse sources, including wastewater, fish and humans, and were from the United States of America, where *A. hydrophila* causes significant economic losses in catfish farming[66].

Despite being considered an emerging human and animal pathogen linked to a wide array of diseases, there is limited available genomic data for the *Aeromonas* genus, particularly from environmental sources. Our core gene phylogeny highlights that this genus is remarkably diverse; this was particularly evident from the high number of novel MLST STs (905), with 1343 (72.47%) genomes assigned to these previously unreported profiles (see Supplementary Data 1). Perhaps unsurprisingly, given the extent of the previously unknown diversity, ANI showed the intra-species ANI values for *A. sobria* and *A. veronii* fell below the standard 95% threshold, challenging current boundaries for species delineation and suggesting a need for reclassification within these species.

Looking at the species distribution between human and environmental samples, from previous studies of children with moderate-to-severe diarrhoea in Karachi, Pakistan, *A. caviae* (64.2%) was seen to be the most numerous, followed by *A. veronii* (19.2%), *A. dhakensis* (9.8%), and *A. enteropelogenes* (4.9%)[46]. Other studies have also shown that *A. hydrophila* is also common in causing intestinal and extra-intestinal infections[67]. From our data, comparing diarrhoea-associated clinical and environmentally derived isolates from South Asia, we showed a similar distribution across both human and environmental samples, with *A. caviae*, *A. veronii*, *A. dhakensis*, and *A. enteropelogenes* being the most abundant.

To identify signatures of intra- and interspecific ecological separation within and between clinical and environmental isolates, we examined all genomes for differences in virulence potential and AMR profiles, linked to taxonomic affiliation and source of isolation. From the virulence gene profiles, there were no significant genetic differences between clinical and environmental strains. However, these profiles allowed us to observe distinct clusters formed by species such as *A. hydrophila*, *A. veronii*, *A. dhakensis*, and *A. salmonicida*, implying that virulence gene profiles are strongly associated with species-level taxonomy, and could be used to classify *Aeromonas* spp. into different groups, serving as a useful marker. For instance, the nearly exclusive presence of *hlyA* in *A. hydrophila* and *A. dhakensis*, as well as the conserved presence of *ela* and *lip* in multiple species, could inform the development of molecular assays for rapid species identification. Furthermore, the consistent clustering patterns observed in PCA and hierarchical analyses highlight the potential of virulence gene profiling as a complementary approach to traditional phylogenetic or multi-locus sequence typing methods.

Most *Aeromonas* genomes, irrespective of source, carried AMR genes, particularly those conferring resistance to β-lactams, aminoglycosides, and sulphonamides, consistent with previous clinical studies[46,65]. Notably, in our study, we identified a wide range of β-lactamase genes, highlighting the problem of widespread β-lactam resistance, consistent with global trends[42,46,68]. These included chromosomally encoded class B metallo-β-lactamases ($bla_{CphA}$), class C cephalosporinases (AmpC β-lactamases: $bla_{FOX}$, $bla_{MOX}$), and class D β-lactamases (oxacillinases: $bla_{OXA}$ variants), underscoring the well-documented intrinsic resistance mechanisms of *Aeromonas* spp.[43,69,70].

Additionally, a notable proportion of isolates carried acquired ESBLs ($n = 52$, 2.8%; e.g., $bla_{CTX-M}$, $bla_{GES}$, $bla_{PER}$, and $bla_{VEB}$) and carbapenemases ($n = 16$, 0.9%; $bla_{KPC}$, $bla_{VIM}$)[71,72]. This is particularly concerning given the lack of standardised treatment guidelines for *Aeromonas* infections, where fluoroquinolones, particularly ciprofloxacin and, in some cases, third-generation cephalosporins are commonly used empirically[32,34,73–75]. Our analysis showed considerable overlap in AMR gene profiles between clinical and environmental isolates, with few genes showing distinct patterns. For example, $bla_{OXA-427}$, a class D carbapenemase previously identified in several *Enterobacteriaceae* clinical strains[76], appeared more frequently in clinical *Aeromonas* isolates. In contrast, The class D carbapenemases gene $bla_{OXA-12}$, first identified in *A. jandaei*[77] and most abundant in environmental isolates, was found in all *A. jandaei*, 565 *A. caviae*, and two *A. hydrophila* isolates, as well as in less common species like *A. allosaccharophila* ($n = 5$), *A. australiensis* ($n = 1$), *A. fluvialis* ($n = 1$), and *A. sobria* ($n = 6$). The gene $bla_{OXA-12}$ was widely distributed across several countries and has been previously reported in environmental *A. veronii* from the United States[78] and in strains from humans and animals in India[79]. The presence of these genes in both clinical and environmental isolates raises concern, as carbapenems are critical last-resort antibiotics[80,81].

We also observed the presence of mobile colistin resistance genes in eight different *Aeromonas* species, spanning diverse geographical regions including Bangladesh, China, Denmark, India, Pakistan, Spain, the United Kingdom, and the United States. Colistin remains a last-resort antibiotic for treating clinical infections caused by multidrug-resistant Gram-negative bacteria[82]. The presence of *mcr* genes, many of which are plasmid-encoded, raises significant public health concerns, as they can facilitate horizontal gene transfer to other clinically important pathogens[82]. The widespread occurrence of *mcr* in both clinical and environmental *Aeromonas*[83–85] isolates further suggests that this genus may act as both a reservoir and conduit for global colistin resistance dissemination.

Notably, the AMR patterns observed in *Aeromonas* spp. from South Asia highlight significant regional variability, with clinical isolates from Pakistan (collected from individuals with moderate to severe diarrhoea)[46] exhibiting resistance across a broader range of antibiotic classes compared to environmental isolates from India and Bangladesh. Nevertheless, environmental isolates often harbour a greater diversity of unique AMR genes within several species, underscoring their role as reservoirs of antimicrobial resistance genes and that antimicrobial contamination of the environment is likely driving their selection and ultimately evolution. Whilst it is clear that many resistance genes are shared between clinical and environmental isolates, with higher proportions of these genes in clinical isolates, it is important to note that the lack of both environmental and clinical isolates from the same geographic region is a limitation of this study. It is also important to note that we lacked phenotypic resistance data; therefore, our findings are based purely on genotypic inference. Overall, this suggests that while environmental isolates contribute to the overall AMR gene pool, selective pressures in clinical settings likely drive the greater prevalence of resistance genes in clinical isolates.

Taken together, the widespread occurrence of clinically relevant resistance genes in *Aeromonas*, a genus found in aquatic and wastewater environments, highlights its role as a key environmental sentinel in the One Health context. This aligns with the broader understanding that various environmental sources contaminated with residual antimicrobial substances, such as wastewater, agricultural runoff, and pharmaceutical discharges[86–88], select for AMR in important bacteria such *Aeromonas* and members of the *Enterobacteriaceae*. These contaminated environments not only support the survival of resistant organisms but also promote their genetic exchange and adaptation, potentially accelerating the spread of resistance across both clinical and environmental settings and harbouring genes conferring resistance to first and last-line antimicrobials.

One of the drivers for this study was that *Aeromonas* spp. are frequently misidentified as *V. cholerae* in clinical and environmental samples[3,89]. Our data shows that 67.15% of the environmental isolates cultured on TCBS in Northern India were *Aeromonas* spp. and not *V. cholerae*. Similarly, of the environmental isolates that were originally collected and identified as *V. cholerae* through culture from the Dhaka household study, ~70% were subsequently reclassified as *Aeromonas* spp. This contrasts with the prevalence of this genus among the 239 stool samples positive for *V. cholerae* from the same study, where only two *Aeromonas* spp. were misidentified as *V. cholerae*, both of which were cultured from asymptomatic individuals, Dr Munir Alam, personal communication (ref. 49, this study). This suggests that misidentification of *Aeromonas* for *V. cholerae* is unlikely to overestimate cholera prevalence in clinical settings (e.g., from human stool). Still, environmental sampling and identification based on microbiological growth and appearance on selective media may overinflate apparent *V. cholerae* prevalence in the environment, especially during a cholera epidemic, if culture is the only confirmatory method employed.

In conclusion, our data brings together the different ecological views of this bacterium; it shows the environment to be an important reservoir of a highly diverse range of species capable of overspilling into humans and, in some instances, causing disease. It is also clear that they carry an important array of therapeutically relevant AMR genes. Here, by taking a broader approach to understanding questions such as sources and sinks of AMR and considering non-traditional organisms, we show that *Aeromonas* has seemingly been hidden in plain sight – both as an emerging human pathogen and clearly as an important environmental reservoir of AMR.

## Methods

### *Aeromonas* isolates and genome sequences

We used a collection of 996 novel *Aeromonas* isolates. Of these, 198 had been originally cultured and identified as *V. cholerae* using conventional microbiological and biochemical techniques[90] before they were sequenced and classified genomically as *Aeromonas*. These included 133 isolates from a study entitled the Cholera-Hospital-Based-Intervention-for-7-Days (ChoBI7) programme in Dhaka city[49] and 65 isolates from an environmental surveillance study for *V. cholerae* conducted in Bangladesh. These 198 genomes were obtained from diverse sources, including drinking water, plankton, zooplankton, sediment, and human stool or rectal swabs, collected between 2004 and 2016. The remaining 798 genomes were obtained from 408 water samples collected for isolation of *V. cholerae* from ponds, rivers, and canals across six different states—Delhi, Haryana, Himachal Pradesh, Punjab, Uttar Pradesh and Uttarakhand, as well as Chandigarh in Northern India—between 2020 and 2023. The collected water samples were first enriched in alkaline peptone water and Selenite F broth. They were then sub-cultured onto various media. Blood agar and Thiosulphate Citrate Bile Salt Sucrose (TCBS; Difco Laboratories, Detroit, Michigan, USA) agar were used for the isolation of *V. cholerae* and *Aeromonas* spp. MacConkey agar (HiMedia Laboratories Private Limited, Mumbai, India) and Xylose Lysine Deoxycholate (XLD; Difco Laboratories, Detroit, Michigan, USA) were used for isolating other *Enterobacteriaceae*. The presumptively selected colonies from the media listed above were identified by matrix-assisted laser desorption ionisation-time of flight mass spectrometry (MALDI-TOF)[91], and isolates that were identified as *Aeromonas* spp. underwent DNA extraction and whole genome sequencing as described below. Between 1 and 9 colonies were retained from each *Aeromonas*-positive water sample in India for downstream processing, including DNA extraction and whole genome sequencing.

DNA was extracted from bacterial isolates using the Wizard® Genomic DNA Purification Kit (Promega, Madison, WI, USA) and Qiagen DNeasy Blood & Tissue Kits (Qiagen, Hilden, Germany). Illumina sequencing libraries were prepared using 0.5 μg DNA and sheared using an S2 ultrasonicator (Covaris), before adaptor ligation and dual index barcoding (NEBNext UltraII, New England Biolabs)[92]. Libraries were sequenced on the HiSeq X10 and NovaSeq 6000 platform (Illumina, San Diego, CA, USA) at the Wellcome Sanger Institute (Cambridge, UK), generating 150 bp paired-end reads with a mean depth of coverage of 68× per genome (range: 21–118×, excluding outliers).

A total of 1853 genomes were analysed, including 996 genomes sequenced in this study. These genomes were contextualised using 416 *Aeromonas* genomes (including one genome from India) downloaded from a library of 661,405 previously assembled bacterial genomes held in the European Nucleotide Archive (ENA)[50], along with 441 genomes from a study on children with moderate to severe diarrhoea and corresponding controls in Pakistan[46].

### Quality control and genome assembly

We performed initial quality control of sequencing reads using FastQC (v0.11.4.)[93] and MultiQC (v1.17)[94]. To further screen for contamination in the raw sequencing reads, we used Kraken2 (v2.2)[95] and Bracken (v2.6.2)[96] (Standard RefSeq database containing archaea, bacteria, viruses, plasmids and human, downloaded from https://benlangmead.github.io/aws-indexes/k2 on 9th October 2023). No trimming of raw reads was performed prior to de novo assembly using SPAdes (v3.9.0)[97] to obtain contiguous scaffolded assemblies. Additional quality control was performed on the de novo assemblies using Quast (v5.0.2)[98] and CheckM (v1.1.2)[99] to calculate the genome contamination and completeness. Assemblies were filtered to retain high-quality assemblies based on genome length (4.5-5.2 Mb), genome contamination (<5%) and genome completeness (>95%). Overall, 5% (97/1950) of the genomes were excluded from the study based on these criteria, including six genomes from the study in Pakistan.

### Species identification and distinction

For species-level classification, we employed GTDB-Tk (v. 2.4.1)[100] with the Genome Taxonomy Database (GTDB; R226, released on 16 April 2025), see Supplementary Data 2. GTDB-Tk integrates marker gene analysis, phylogenetic placement, and average nucleotide identity (ANI) to provide robust and consistent taxonomic assignments. This was followed by assessment of genome-relatedness indices, where we used FastANI (v1.3)[101] for the pairwise comparison of the *Aeromonas* genomes. For a subset of genomes ($n = 49$), digital DNA−DNA hybridisation (dDDH) assays were also performed to further refine species-level resolution (Supplementary Data 3). The FastANI threshold was set at 90% based on pairwise sequence mapping[101], and heatmaps based on estimated ANI for all *Aeromonas* genomes were created using the Complex heatmap package in R (v4.4.3)[102].

### Prediction of AMR and virulence genes

AMR genes were identified from the assembled contigs using ABRicate (v1.0.1, available at https://github.com/tseemann/abricate) to query the Comprehensive Antimicrobial Resistance Database (CARD, v3.2.4, last updated in August 2022)[103], where the CARD identification threshold was set at ≥80% and <80% nucleotide identity, both with ≥80% coverage (Supplementary Data 1). For the identification of the putative virulence genes, we initially used ABRicate with the Virulence Finder Database[104], but this did not yield any hits. We subsequently used in_silico_pcr (v1.0.0, available at https://github.com/sangerpathogens/sh16_scripts/blob/master/legacy/in_silico_pcr.py) with previously described primer sequences (see Supplementary Table 8)[105,106] to detect virulence genes. To explore the patterns of AMR and virulence gene distribution in relation to sample origin (clinical vs. environmental) and *Aeromonas* species, we conducted a Principal Component Analysis (PCA) using the prcomp function from the stats package in R (v4.4.3)[102], based on presence/absence matrices. The first two components, PCA1 and PCA2, were used. Furthermore, we applied Non-metric Multidimensional Scaling (NMDS) with the metaMDS

function from the vegan package in R (v4.4.3)[102], to investigate whether the AMR gene diversity of sample clusters based on the species or the source of isolation (clinical and environmental), using Bray-Curtis dissimilarity to achieve higher resolution. The two-dimensional ordinations from both PCA and NMDS were visualised using the ggplot2 package.

## Genome annotation and phylogenetic analysis

Draft *Aeromonas* genomes were annotated using PROKKA (v1.4.5)[107] with default parameters. We used Panaroo (v1.3.0)[108] to infer the pangenome of 1853 genomes and used the resulting core gene alignment (comprising 1969 core genes) for phylogenetic analysis. We extracted 1,282,613 variable sites using SNP-sites (v2.5.1)[109], and the resulting multiple sequence alignment file was used to infer a phylogeny with IQ-TREE (v2.1)[110], using a generalised time reversible substitution model with a FreeRate model of heterogeneity[111], incorporating invariant sites using the -fconst parameter, and with 1000 UltraFast Bootstraps[112]. Phylogenetic trees and associated metadata were visualised in iTOL (v.5.6)[113]. Similarly, a separate phylogenetic tree was constructed for genomes from India, Bangladesh, and Pakistan (*n* = 1438), consisting of 2067 core genes and 1,197,118 variable sites.

## Multi-locus sequence typing (MLST) and subtyping of *Aeromonas* isolates

We used the MLST tool (v2.9, available at https://github.com/tseemann/mlst) to infer MLST against the *Aeromonas* scheme developed by Martino and colleagues[51] and hosted at https://pubmlst.org/organisms/aeromonas-spp[114]. This MLST scheme uses six housekeeping genes (*gyrB*, *groL*, *gltA*, *metG*, *ppsA*, and *recA*) to subtype *Aeromonas* below the species level into Sequence Types (STs). Novel alleles and STs were submitted to the *Aeromonas* MLST database via pubmlst.org for naming.

Given the broad distribution of genomes across diverse MLST types among species, we also employed Bayesian Analysis of Population Structure (BAPS) using fastBAPS (v1.0.6)[115] to define higher-level groupings.

## Reporting summary

Further information on research design is available in the Nature Portfolio Reporting Summary linked to this article.

## Data availability

Short-read sequence data (150 bp paired-end reads) are available in the EMBL Nucleotide Sequence Database (European Nucleotide Archive, http://www.ebi.ac.uk/ena); accession numbers are listed in Supplementary Data 1. Additional data supporting the findings of this study are available in the Supplementary Information. Additional data supporting the findings of this study, including R scripts and documented command-line workflows for genomic analyses, are available at Figshare (v4, https://doi.org/10.6084/m9.figshare.30833858).

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

## Acknowledgements

This research was funded in whole, or in part, by the Wellcome Trust [to NS, ROG, MAB, MJD, AC, NRT: Grant #206194, 220540/Z/20/A; CT, NT, MA, NRT: Grant 215704/Z/19/Z]. This research was also supported by the US National Institutes of Health (Grant 1R01AI3912901) and by USAID (Grant AID-OAA-F-15-00038) [to FTJ, MR, SM, FZ, TP, SIB, MS, MA]. For the purpose of open access, the authors have applied a CC-BY public copyright licence to any author-accepted manuscript version arising from this submission. The authors thank the teams involved in sample collection in Bangladesh and India, as well as the Parasites and Microbes Programme Sequencing and Informatics teams at the Wellcome Sanger Institute. We also thank the National Biodiversity Authority of India. MA acknowledges with gratitude the contribution of collaborating colleagues involved in the grant in Bangladesh and the USA, including the laboratory and field team members of icddr,b. icddr,b acknowledges the donors who provide unrestricted support for its operations and research.

## Author contributions

Conceptualisation: N.S., R.O.G., M.A.B., M.J.D., A.C., A.T.B., N.R.T.; Formal analysis: N.S., R.O.G., M.A.B., N.R.T.; Investigation: N.S., C.T., F.T.J., M.R., S.M., F.Z., T.P., S.I.B., M.S., M.A., N.T., N.R.T.; Resources: M.A., N.T., N.R.T.; Writing and Visualisation: N.S., R.O.G., M.A.B., N.R.T.; Supervision: M.A.B., M.J.D., A.C., A.T.B., S.K., N.T., M.A., N.R.T.; Review and Editing: N.S., R.O.G., C.T., M.A.B., M.J.D., A.C., A.T.B., F.T.J., M.R., S.M., F.Z., T.P., S.I.B., M.S., B.M., D.D., C.M.G., S.K., M.A., N.T., N.R.T. All authors reviewed the manuscript.

## Competing interests

The authors declare no competing interests.
