## [Transparent Peer Review file · Nature Communications]

Aeromonas in South Asia: Genomic Insights into an Environmental Pathogen and Reservoir of Antimicrobial Resistance

Corresponding Author: Professor Nicholas Thomson

Version 0:

Reviewer comments:

Reviewer #1

(Remarks to the Author)

This manuscript investigates the genomic characteristics of *Aeromonas* spp. using a broad collection of isolates from South Asia. Although the dataset is extensive, the study contains several methodological and interpretative deficiencies that significantly limit the validity of its conclusions and should be thoroughly addressed before publication.

The Introduction section should not solely focus on human infections. *Aeromonas* spp. are also capable of causing infections in terrestrial and aquatic animals; this aspect should be elaborated. Moreover, the zoonotic potential of these species should be emphasized.

A dedicated paragraph discussing the diagnostic challenges associated with *Aeromonas* identification and methods for achieving definitive diagnosis should be included in the Introduction.

Additional details should be provided for the isolates mentioned on lines 142–143, including the time frame of isolation and specific sources.

On line 442, the number of isolates subjected to genome analysis should be explicitly stated.

Although the WGS platform (HiSeq X10) is mentioned, essential methodological details such as library preparation procedures, targeted sequencing depth, and genome coverage are lacking and should be added.

The methodology does not clearly indicate whether raw reads were subjected to trimming or quality filtering post-NGS. This information should be clarified.

In the genome-based identification section, it would be more appropriate to use DNA-DNA hybridization (dDDH) values instead of Kraken and Bracken. dDDH, especially when combined with ANI values, provides a more reliable taxonomic resolution.

The database used for species identification is from 2023. However, even within the current year, several new *Aeromonas* species have been described. The study should ensure the use of the most recent taxonomic references.

In the section on prediction of AMR and virulence genes, the threshold values (identity and coverage) should be clearly stated. Additionally, the dates of the last database updates should be provided.

The PubMLST link cited on line 497 is incorrect. It should be updated to the correct URL: "<https://pubmlst.org/organisms/aeromonas-spp>". The section titled "Multi-locus Sequence Typing (MLST) and Subtyping of *Aeromonas* isolates" should be revised accordingly.

The Results section lacks proper evaluation due to incorrect species assignments, which compromises the interpretation of findings.

Reviewer #2

(Remarks to the Author)

The manuscript presents a comprehensive view of the *Aeromonas* genus, including 1,853 genomes, and a detailed comparison of clinical and environmental strains from South Asia, with 996 newly sequenced genomes from Bangladesh and India. The authors have identified a high diversity of *Aeromonas* without a clear separation between clinical and environmental isolates. They also observed a high incidence of AMR genes across all isolates, including resistance to front-line and last-line antibiotics. Additionally, the manuscript highlights the frequent misidentification of *Aeromonas* as *Vibrio cholerae*, which is a key issue in cholera-endemic regions where both genera coexist and are associated with diarrheal disease. The findings are of interest to readers, but the representativeness of the dataset and the depth of the analysis are insufficient. Below are my comments:

1. The title should be revised to explicitly state the geographic focus of the study, as the dataset primarily includes strains from South Asia. This is important for clarity and to avoid misleading readers about the scope of the research. Additionally, in Figure 1, the species *Aeromonas salmonicida* appears to be represented by nearly identical clones, which raises questions about the representativeness of the sampled strains. This might be a limitation in the study's ability to reflect the true genetic diversity of the genus.
2. The authors only searched the ENR database for genome data, but the NCBI database currently contains over 3,000 *Aeromonas* genomes. Including these additional genomes would provide a more comprehensive view of the genus and enhance the understanding of the public health significance of *Aeromonas* in terms of AMR dissemination and global transmission dynamics.
3. In the introduction section, it is recommended to appropriately introduce the current status of antibiotic resistance in *Aeromonas* species to enhance the reader's understanding. For example, existing studies have shown that *Aeromonas* is a natural host for MCR-3 and MCR-7, [10.1016/j.drug.2024.101088] [10.1016/j.drug.2023.101006].
4. The manuscript's genome analysis lacks sufficient depth and accuracy, which limits the ability to draw clear conclusions. For example, the identification threshold for resistance genes was set at 80% nucleotide identity, but many studies have shown that resistance genes such as *mcr-7* in *Aeromonas* often have lower nucleotide identity. Additionally, many resistance genes are naturally carried by *Aeromonas*. Whether these genes can evolve into acquired drug resistance genes and spread remains to be further analyzed.
5. It is recommended that the authors supplement the manuscript with data on the phenotypic drug resistance of the strains. This would help clarify the contribution of the identified resistance genes to the overall drug resistance profile of the isolates.

Reviewer #3

(Remarks to the Author)

Synopsis

This study conducted environmental surveillance of *Aeromonas* spp. in India and Bangladesh and performed WGS on 996 isolates. They then combined these newly sequenced data with publicly available genomes of 441 clinical *Aeromonas* isolates from Pakistan, as well as a global collection of 416 *Aeromonas* isolates from different sources. A set of bioinformatic tools, including Kraken 2, Bracken, GTDB-Tk and FastANI were used to determine *Aeromonas* species. Standard genomic and phylogenetic tools were employed to define the population genetic structure and characterize the distribution of virulence and AMR genes by species and sample type.

They found a highly diverse *Aeromonas* genus with 28 species and 905 novel sequence types (STs), comprising 72.5% of total genomes. Clinical and environmental isolates clustered together across the phylogenetic tree. A high prevalence of AMR genes was found across isolates, including predicted resistance to last-line antibiotics. They further highlighted a frequent misidentification of *Aeromonas* as *Vibrio cholerae* in cholera endemic settings. They concluded that *Aeromonas* is an important environmental AMR reservoir, encompassing multi-species capable of causing human infections.

Strengths:

There has been a paucity of genomic data on *Aeromonas*, particularly in South Asian countries, where little is known about its environmental circulation and potential for spillover into human disease, such as childhood diarrhea. This study presents large-scale genomic surveillance of environmental isolates, offering new insights into the high species and genetic diversity of *Aeromonas*, its role as a major AMR reservoir, including resistance to last-resort antibiotics, and the strong genetic links between environmental and diarrhea-causing human isolates. Within this genomic framework, the authors have contributed to the expansion of the *Aeromonas* MLST database, with 905 novel sequence types now defined, providing a valuable resource for the broader research community. Notably, the high co-occurrence of *Aeromonas* and *Vibrio cholerae* in environmental samples is an important finding, raising concerns about the potential for overestimation of cholera prevalence in environmental surveillance, particularly in endemic settings.

Limitations:

Because the authors compared species, sequence types, AMR, and virulence gene distributions between clinical isolates (sourced from Pakistan) and environmental isolates (from Bangladesh and Pakistan), and given that the sampling strategies differed between India (multiple colonies per sample) and Bangladesh (single colony per sample), the manuscript should include a paragraph discussing potential biases introduced by these differences in sampling approach, geographical coverage, and isolate sources. Additionally, the study would be strengthened by providing more detailed metadata regarding the environmental water sources; specifically, whether the samples were from drinking water, rivers, lakes, or streams. Information such as sampling locations, presence of antibiotic residues, linkage to human activity, and whether repeat sampling at the same sites occurred (if available) should be included.

Throughout the manuscript, the authors use the term “clinical and environmental isolates.” It appears that this refers only to clinical isolates from Pakistan and environmental isolates from India and Bangladesh, excluding the global isolate collection. This definition should be made explicit to avoid confusion. Moreover, numerical values should be provided alongside all percentages mentioned in the manuscript to enhance clarity and interpretability.

Specific comments and questions:

- Line 162: Please provide more details on the global isolates, including their sources (e.g., environment, human disease, animal disease, human carriage), whether they were associated with outbreaks, and their years of isolation.
- Line 181: Consider rewriting and splitting this sentence for clarity.
- Line 187: Provide the number of samples containing more than one species.
- Line 188: Without associated metadata, interpreting country-specific species distribution from the global dataset is difficult. Please acknowledge this limitation or provide metadata.
- Line 199: Do the authors have co-occurrence data for the Bangladesh site?
- Line 207: Add a separate paragraph that describes in detail the phylogenetic relatedness of environmental and clinical isolates, particularly for species and specific sequence types where clusters are evident.
- Line 229: Clarify the phrase “human health.” Also, consider to revise the header to specify “Virulence Distribution.”
- Line 231: Add citations for each virulence factor mentioned.
- Line 235: Can the authors explain why ABRicate with VFDB did not detect any hits? Did they attempt alternative tools like SRST2?
- Line 239: The color coding in Figure 2a does not match the legend, please revise.
- Line 240: Split the sentence into two and add more detail about the “virulence profile.”
- Line 243: Specify what is meant by “others.”
- Line 246: Clarify what is meant by “highest proportion of some virulence genes.” Which genes and which species?
- Line 250: Add percentages where relevant.
- Lines 251–253: Provide the proportions of hlyA and ast across species. If these genes are proposed as species markers, this information is essential.
- Line 254: This sentence should be rewritten as two sentences for clarity.
- Lines 260–265: The association between species and sample origin appears to have already been discussed in line 178—please check for redundancy.
- Line 266: This content would be better placed under the “Genomic Taxonomy” section header.
- Line 274: Provide percentages for each β -lactam antibiotic subgroup.
- Line 276: Provide the proportion of mcr genes found in each of the eight species. Were these isolates phenotypically tested for colistin resistance?
- Line 282: It seems that the authors are comparing AMR gene distributions across countries, not phenotypic resistance, please clarify. Note that the correlation between AMR genotype and phenotype is not well-studied in these organisms.
- Lines 291–296: These sentences would be better placed in the Discussion and expanded upon.
- Line 298: Consider including comparisons of AMR gene profiles between clinical and environmental isolates for the predominant species.
- Line 324: Consider adding more references (e.g., PMID: 38739115; PMID: 37808293).
- Line 326: According to Supplementary Figure 1B, *A. dhakensis* may not be overrepresented in Bangladesh, please clarify.
- Lines 327–331: Metadata for global isolates are needed to understand their epidemiological context.
- Line 346: Is this line referring to diarrheal disease? In PMID: 38739115, *A. dhakensis* appear to be more prevalent than other *Aeromonas* species in human bloodstream infections.
- Line 353: Expand on the interpretation of the presented genetic data.
- Line 354: Please provide further clarification, particularly if this information is intended to inform the development of future assays for *Aeromonas* species classification
- Line 364: Confirm whether a substantial number of ESBL genes were identified.
- Line 365: Replace with “AMR gene profile.”
- Lines 365–374: Differences in AMR and virulence gene distribution may reflect the fact that all clinical isolates were from Pakistan, while environmental isolates were more diverse and came from two countries. This should be discussed.
- Line 387: Please provide more explanation on why isolates from Pakistan harbored a higher number of resistance genes. Could this be attributed to their clinical origin, where selection pressure from antibiotic use is typically higher? Alternatively, could the difference be influenced by the absence of environmental isolates from Pakistan in the study, limiting a direct comparison across sample types?
- Line 427: Please provide additional metadata for the isolates from Bangladesh.
- Line 501: This sentence is difficult to understand, please rewrite for clarity.
- Line 807: Consider revising to: “Each dot is coloured based on the gene abundance per species (see scale), and the size of the dots corresponds to the number of genomes harboring the gene.”

Version 1:

Reviewer comments:

Reviewer #1

(Remarks to the Author)

Dear Authors,

Thank you for the substantial and carefully executed revision of “Aeromonas in South Asia: Genomic Insights into an Environmental Pathogen and Reservoir of Antimicrobial Resistance.” I appreciate the scope of the study and the evident effort of the team, and I offer the following comments purely in a constructive spirit.

My remaining concern is the taxonomic identification of the *Aeromonas* isolates. Based on my research and experience with *Aeromonas*, average nucleotide identity (ANI) alone is not sufficient for species-level assignments, particularly in borderline cases. I strongly recommend that you combine ANI with digital DNA–DNA hybridization (dDDH), reporting both metrics and using accepted thresholds (e.g., ANI \approx 95–96% and dDDH \approx 70%) to reach stable conclusions. Please provide a table (preferably supplementary) listing, for each isolate, the closest type strain(s) together with ANI and dDDH values and the corresponding genome accessions, along with the software and parameters used (e.g., FastANI/pyani for ANI; GGDC/Formula 2 for dDDH).

Relatedly, while GTDB/GTDB-Tk is valuable for genome-based taxonomy, it is not a nomenclatural authority and may include provisional or non-validated names and an incomplete set of type strains. In the current GTDB release you cite, I noted ~32 *Aeromonas* taxa irrespective of nomenclatural status, whereas the List of Prokaryotic names with Standing in Nomenclature (LPSN; <https://lpsn.dsmz.de/genus/aeromonas>) currently recognises 34 validly published *Aeromonas* species. Please make sure your reference panel includes the type strain genomes of all valid species listed by LPSN, not only those represented in GTDB. Augmenting the GTDB-Tk reference with missing valid *Aeromonas* type strains may change some assignments, and this possibility should be checked and, if relevant, discussed.

Addressing these points will, I believe, make the taxonomic backbone of the manuscript more robust and increase confidence in the downstream ecological and AMR interpretations. Thank you again for your thorough work; I hope these suggestions are helpful.

Reviewer #2

(Remarks to the Author)

The authors have resolved all my issues.

Reviewer #3

(Remarks to the Author)

I thank the authors for their thorough revision of the manuscript and for addressing my previous comments and questions. The data interpretation and overall clarity have been substantially improved. One minor comment remains: please revise the color scheme and legend of Supplementary Figure 5b to enhance clarity.

Open Access This Peer Review File is licensed under a Creative Commons Attribution 4.0 International License, which permits use, sharing, adaptation, distribution and reproduction in any medium or format, as long as you give appropriate credit to the original author(s) and the source, provide a link to the Creative Commons license, and indicate if changes were

made.

TITLE: *Aeromonas* in South Asia: Genomic Insights into an Environmental Pathogen and Reservoir of Antimicrobial Resistance.

Nature Communications manuscript NCOMMS-25-34237

We are sincerely thankful to the reviewers for their thoughtful and constructive comments. Their perspectives have helped us refine the manuscript in meaningful ways. Below, we provide detailed responses to each point raised, and we have revised the manuscript accordingly.

REVIEWER COMMENTS

Reviewer #1 (Remarks to the Author)

This manuscript investigates the genomic characteristics of *Aeromonas* spp. using a broad collection of isolates from South Asia. Although the dataset is extensive, the study contains several methodological and interpretative deficiencies that significantly limit the validity of its conclusions and should be thoroughly addressed before publication.

1. The Introduction section should not solely focus on human infections. *Aeromonas* spp. are also capable of causing infections in terrestrial and aquatic animals; this aspect should be elaborated. Moreover, the zoonotic potential of these species should be emphasized.

Done. Please see lines 109-114 for the updated text.

2. A dedicated paragraph discussing the diagnostic challenges associated with *Aeromonas* identification and methods for achieving definitive diagnosis should be included in the Introduction.

Done. Please see lines 132-141.

3. Additional details should be provided for the isolates mentioned on lines 142–143, including the time frame of isolation and specific sources.

Done. Please see lines 160-162.

4. On line 442, the number of isolates subjected to genome analysis should be explicitly stated.

Done. Please see line 513.

5. Although the WGS platform (HiSeq X10) is mentioned, essential methodological details such as library preparation procedures, targeted sequencing depth, and genome coverage are lacking and should be added.

Done. Please see lines 508-511.

6. The methodology does not clearly indicate whether raw reads were subjected to trimming or quality filtering post-NGS. This information should be clarified.

To clarify, no trimming of raw reads was performed prior to de novo assembly. In addition, CheckM was also used for quality assurance to exclude incomplete or contaminated assemblies. This is all now detailed in the Methods lines 520-524.

7. In the genome-based identification section, it would be more appropriate to use DNA-DNA hybridization (dDDH) values instead of Kraken and Bracken. dDDH, especially when combined with ANI values, provides a more reliable taxonomic resolution.

We apologise; this was wrongly stated in the Methods (previously line 462). Kraken/Bracken were used only to screen for contamination, not for species delineation (see lines 521–523).

*Species identification was performed using **GTDB-Tk** (previously line 464, now lines 532–535), which provides taxonomic assignments based on the Genome Taxonomy Database (GTDB), a widely recognized reference standard for bacterial and archaeal genomes (Chaumeil et al., 2020, PMID: 31730192). GTDB-Tk integrates average nucleotide identity (ANI), marker gene-based classification, and phylogenetic placement to provide a comprehensive species-level taxonomic assessment.*

To further support species assignments, we applied FastANI, which provides ANI estimates comparable to dDDH (~70% dDDH = ~95% ANI; Goris et al., 2007, PMID: 17220447). Given this concordance and to avoid using multiple overlapping tools, we focused on GTDB-Tk with FastANI for efficient and consistent species assignment in the manuscript.

8. The database used for species identification is from 2023. However, even within the current year, several new *Aeromonas* species have been described. The study should ensure the use of the most recent taxonomic references.

In response to your question, we repeated this analysis with the latest GTDB release (R226, 16 April 2025; see Supplementary Data-2), which did not change the results for species diversity or composition.

9. In the section on prediction of AMR and virulence genes, the threshold values (identity and coverage) should be clearly stated. Additionally, the dates of the last database updates should be provided.

Done. Please see lines 543 and 544.

10. The PubMLST link cited on line 497 is incorrect. It should be updated to the correct URL: "<https://pubmlst.org/organisms/aeromonas-spp> [pubmlst.org]". The section titled "Multi-locus Sequence Typing (MLST) and Subtyping of *Aeromonas* isolates" should be revised accordingly.

Done. Please see lines 242 and 571.

11. The Results section lacks proper evaluation due to incorrect species assignments, which compromises the interpretation of findings.

Please see the responses to points 7 and 8 above. Given this we would argue that the species assignments are robust and therefore so too is the interpretations of the data.

Reviewer #2 (Remarks to the Author):

The manuscript presents a comprehensive view of the *Aeromonas* genus, including 1,853 genomes, and a detailed comparison of clinical and environmental strains from South Asia, with 996 newly sequenced genomes from Bangladesh and India. The authors have identified a high diversity of *Aeromonas* without a clear separation between clinical and environmental isolates. They also observed a high incidence of AMR genes across all isolates, including resistance to front-line and last-line antibiotics. Additionally, the manuscript highlights the frequent misidentification of *Aeromonas* as *Vibrio cholerae*, which is a key issue in cholera-endemic regions where both genera coexist and are associated with diarrheal disease. The findings are of interest to readers, but the representativeness of the dataset and the depth of the analysis are insufficient. Below are my comments:

1. The title should be revised to explicitly state the geographic focus of the study, as the dataset primarily includes strains from South Asia. This is important for clarity and to avoid misleading readers about the scope of the research.

Thank you for your comment this was extremely helpful. As suggested, we have revised the title.

2. Additionally, in Figure 1, the species *Aeromonas salmonicida* appears to be represented by nearly identical clones, which raises questions about the representativeness of the sampled strains. This might be a limitation in the study's ability to reflect the true genetic diversity of the genus.

*The observation from Figure 1 that *Aeromonas salmonicida* appears to be represented by nearly identical clones, we would like to highlight that among the 176 *A. salmonicida* genomes, the majority originate from Denmark (100 genomes, 1980–2011) and China (60 genomes, 2012–2015), covering a broad temporal range. The remaining genomes come from other countries. A comprehensive sampling overview has been added to the revised manuscript to further clarify the dataset's composition and its implications for genetic diversity (see Supplementary Table 1).*

3. The authors only searched the ENR database for genome data, but the NCBI database currently contains over 3,000 *Aeromonas* genomes. Including these additional genomes would provide a more comprehensive view of the genus and enhance the understanding of the public health significance of *Aeromonas* in terms of AMR dissemination and global transmission dynamics.

*Likely because of the increasing significance of *Aeromonas* in human infection, this is a rapidly changing situation with more genomes appearing each month. To address your comment, we retrieved all 4,378 *Aeromonas* genomes from NCBI (as of August 18, 2025) and classified them using GTDB-Tk. After excluding 30 assemblies that could not be assigned at the species level, 4,344 assemblies remained.*

*Overall, the NCBI dataset contained 32 *Aeromonas* species, the majority of which were represented by five species (*A. veronii*, *A. caviae*, *A. hydrophila*, *A. salmonicida*, and *A. dhakensis*), which together accounted for ~86% (3,722/4,344) of genomes. This species distribution is, consistent with our data (~89%, 1,660/1,853). The species present in the NCBI dataset but absent from our study (*A. aquatica*, *A. finlandensis*, *A. lacus*, *A. lusitana*, and *A. molluscorum*) accounted for only ~0.5% of the total NCBI genomes (20/4,344).*

However, NCBI dataset has a strong geographic bias, dominated by genomes from China (1,436), and from East Asia (1,785 genomes; ~41%), please see Figure below. Only 531 of the 4,344 genomes (~12%) originate from South Asia, (most of which are unpublished sequences from Bangladesh). Hence in principle this extra data is not relevant to the geographic focus of this study and given the 531 have only recently been submitted and are unpublished we are reluctant to include them in our publication, in fear of disadvantaging others. However, to address your question and to increase the species and geographic contextual data, without losing our focus on South Asia we incorporated the curated dataset published by Grace Blackwell et al. (PMID: PMC8577725) to our phylogenies, which included 28

Aeromonas species from 16 countries across five continents and diverse sources (see revised Supplementary Table 1).

Figure: Overview of *Aeromonas* genomes available in GenBank (as of August 18, 2025).

4. In the introduction section, it is recommended to appropriately introduce the current status of antibiotic resistance in *Aeromonas* species to enhance the reader's understanding. For example, existing studies have shown that *Aeromonas* is a natural host for MCR-3 and MCR-7, [10.1016/j.drug.2024.101088] [10.1016/j.drug.2023.101006].

Thank you for the suggestion. We have revised the Introduction to include this information. Please see lines 143-151.

5. The manuscript's genome analysis lacks sufficient depth and accuracy, which limits the ability to draw clear conclusions. For example, the identification threshold for resistance genes was set at 80% nucleotide identity, but many studies have shown that resistance genes such as *mcr-7* in *Aeromonas* often have lower nucleotide identity. Additionally, many resistance genes are naturally carried by *Aeromonas*. Whether these genes can evolve into acquired drug resistance genes and spread remains to be further analyzed.

*Thank you for your comment. We reanalysed the data using a more inclusive threshold of $\geq 30\%$ nucleotide identity combined with $\geq 80\%$ coverage. Importantly, below the 80% nucleotide identity threshold, only the following MCR variants were identified: MCR-3, MCR-3.6, MCR-3.8, MCR-3.9, MCR-3.12, MCR-7.1, and MCR-9; no other AMR genes were detected. This information is presented in Supplementary Data-1, under a separate column named “**AMR_Genes (Coverage $\geq 80\%$, Nucleotide Identity $< 80\%$).**”*

6. It is recommended that the authors supplement the manuscript with data on the phenotypic drug resistance of the strains. This would help clarify the contribution of the identified resistance genes to the overall drug resistance profile of the isolates.

*Here, unfortunately phenotypic AST testing was not standard practise or standardised across the multiple different labs collecting these isolates, especially for environmentally derived *Aeromonas* isolates. What is more, we did not have access to the live isolates, hence we could not test them consistently either. Given this and to ensure consistency we chose to focus on the genetic differences, including in antimicrobial resistance gene profiles, between *Aeromonas* isolates from different sample sites and across species. Given the above and whilst these data provide important insights into the resistome and the associated risks to both aquatic environments and human health we would argue that adding this analysis is out of scope for this study. We have now clarified this limitation in the Discussion section (Please see lines 448-449).*

Reviewer #3 (Remarks to the Author)

Synopsis

This study conducted environmental surveillance of *Aeromonas* spp. in India and Bangladesh and performed WGS on 996 isolates. They then combined these newly sequenced data with publicly available genomes of 441 clinical *Aeromonas* isolates from Pakistan, as well as a global collection of 416 *Aeromonas* isolates from different sources. A set of bioinformatic tools, including Kraken 2, Bracken, GTDB-Tk and FastANI were used to determine *Aeromonas* species. Standard genomic and phylogenetic tools were employed to define the population genetic structure and characterize the distribution of virulence and AMR genes by species and sample type.

They found a highly diverse *Aeromonas* genus with 28 species and 905 novel sequence types (STs), comprising 72.5% of total genomes. Clinical and environmental isolates clustered together across the phylogenetic tree. A high prevalence of AMR genes was found across isolates, including predicted resistance to last-line antibiotics. They further highlighted a frequent misidentification of *Aeromonas* as *Vibrio cholera* in cholera endemic settings. They concluded that *Aeromonas* is an important environmental AMR reservoir, encompassing multi-species capable of causing human infections.

Strengths:

There has been a paucity of genomic data on *Aeromonas*, particularly in South Asian countries, where little is known about its environmental circulation and potential for spillover into human disease, such as childhood diarrhea. This study presents large-scale genomic surveillance of environmental isolates, offering new insights into the high species and genetic diversity of *Aeromonas*, its role as a major AMR reservoir, including resistance to last-resort antibiotics, and the strong genetic links between environmental and diarrhea-causing human isolates. Within this genomic framework, the authors have contributed to the expansion of the *Aeromonas* MLST database, with 905 novel sequence types now defined, providing a valuable resource for the broader research community. Notably, the high co-occurrence of *Aeromonas* and *Vibrio cholerae* in environmental samples is an important finding, raising concerns about the potential for overestimation of cholera prevalence in environmental surveillance, particularly in endemic settings.

Limitations:

Because the authors compared species, sequence types, AMR, and virulence gene distributions between clinical isolates (sourced from Pakistan) and environmental isolates (from Bangladesh and Pakistan), and given that the sampling strategies differed between India (multiple colonies per sample) and Bangladesh (single colony per sample), the manuscript should include a paragraph discussing potential biases introduced by these differences in sampling approach, geographical coverage, and isolate sources. Additionally, the study would be strengthened by providing more detailed metadata regarding the environmental water sources; specifically, whether the samples were from drinking water, rivers, lakes, or streams. Information such as sampling locations, presence of antibiotic residues, linkage to human activity, and whether repeat sampling at the same sites occurred (if available) should be included. Throughout the manuscript, the authors use the term “clinical and environmental isolates.” It appears that this refers only to clinical isolates from Pakistan and environmental isolates from India and Bangladesh, excluding the global isolate collection. This definition should be made explicit to avoid confusion. Moreover, numerical values should be provided alongside all percentages mentioned in the manuscript to enhance clarity and interpretability.

Specific comments and questions:

1. Line 162: Please provide more details on the global isolates, including their sources (e.g., environment, human disease, animal disease, human carriage), whether they were associated with outbreaks, and their years of isolation.

Thank you for your comment. We have included the available metadata for the global isolates in the revised manuscript. Please see lines 181-183 and Supplementary Table 1.

2. Line 181: Consider rewriting and splitting this sentence for clarity.

Done. Please see lines 201-204.

3. Line 187: Provide the number of samples containing more than one species.

Done. Please see line 206 and updated Supplementary Fig. 3a.

4. Line 188: Without associated metadata, interpreting country-specific species distribution from the global dataset is difficult. Please acknowledge this limitation or provide metadata.

Done. To address this comment Metadata has been included in the revised manuscript (please see Supplementary Table 1).

5. Line 199: Do the authors have co-occurrence data for the Bangladesh site?

Unfortunately, this is not available.

6. Line 207: Add a separate paragraph that describes in detail the phylogenetic relatedness of environmental and clinical isolates, particularly for species and specific sequence types where clusters are evident.

Added, please see lines 296-298 and revised Supplementary Table 3.

7. Line 229: Clarify the phrase "human health." Also, consider to revise the header to specify "Virulence Distribution."

Done. We have revised the header to explicitly reflect the focus on virulence gene distribution (please see line 250).

8. Line 231: Add citations for each virulence factor mentioned.

Added, please see lines 252-255.

9. Line 235: Can the authors explain why ABRicate with VFDB did not detect any hits? Did they attempt alternative tools like SRST2?

*We had not used SRST2 in our original analysis. However, to address your comment, we ran SRST2 and similarly found no virulence gene matches. ABRicate with VFDB did not detect any virulence genes because the VFDB database lacks curated entries specific to *Aeromonas* spp., focusing mainly on Enterobacteriaceae and other well-characterized pathogens. Thus, it is clear that the absence of hits is due to the database content rather than the tool used. To address this limitation, we conducted a thorough literature review to identify experimentally validated *Aeromonas*-associated virulence genes. We then compiled a custom gene set and used these data as a bespoke database with which to identify all known virulence genes across our genomes using in-silico PCR.*

10. Line 239: The color coding in Figure 2a does not match the legend, please revise.

Done.

11. Line 240: Split the sentence into two and add more detail about the “virulence profile”.

Done. Additional details about the “virulence profile” have been included (please see lines 260-269).

12. Line 243: Specify what is meant by “others”.

Added, please see lines 267.

13. Line 246: Clarify what is meant by “highest proportion of some virulence genes.” Which genes and which species?

Done. Please see lines 269-273.

14. Line 250: Add percentages where relevant.

Done.

15. Lines 251–253: Provide the proportions of hlyA and ast across species. If these genes are proposed as species markers, this information is essential.

Done.

16. Line 254: This sentence should be rewritten as two sentences for clarity.

Done. Please see lines 281-285.

17. Lines 260–265: The association between species and sample origin appears to have already been discussed in line 178—please check for redundancy.

Thank you, the difference is that lines 260–265 (now lines 287–294) refer to the association between virulence gene presence and sample of origin.

18. Line 266: This content would be better placed under the “Genomic Taxonomy” section header.

Thank you, we have revised the section for clarity.

19. Line 274: Provide percentages for each β -lactam antibiotic subgroup.

Done. Please see lines 306-312.

20. Line 276: Provide the proportion of mcr genes found in each of the eight species. Were these isolates phenotypically tested for colistin resistance?

We have updated the section to include mcr gene distribution per species. To clarify these isolates were not phenotypically tested. Please also see response to Reviewer 2, question 6.

21. Line 282: It seems that the authors are comparing AMR gene distributions across countries, not phenotypic resistance, please clarify. Note that the correlation between AMR genotype and phenotype is not well-studied in these organisms.

Done.

22. Lines 291–296: These sentences would be better placed in the Discussion and expanded upon.

Done. Please see lines 413-420.

23. Line 298: Consider including comparisons of AMR gene profiles between clinical and environmental isolates for the predominant species.

To address this, we have provided detailed AMR gene information for clinical and environmental isolates by species (please see lines 335-348, Supplementary Figures 7 and 8, and Supplementary Table 6 and 7).

24. Line 324: Consider adding more references (e.g., PMID: 38739115; PMID: 37808293).

Added.

25. Line 326: According to Supplementary Figure 1B, *A. dhakensis* may not be overpresented in Bangladesh, please clarify.

Corrected, thank you for pointing this out.

26. Lines 327–331: Metadata for global isolates are needed to understand their epidemiological context.

Added, please see lines 373-378.

27. Line 346: Is this line referring to diarrheal disease? In PMID: 38739115, *A. dhakensis* appear to be more prevalent than other *Aeromonas* species in human bloodstream infections.

Thank you for your comment. Yes, in line 346 we are referring to diarrheal cases, not bloodstream infections. We had revised the text for clarity.

28. Line 353: Expand on the interpretation of the presented genetic data.

Done, please see lines 397-403.

29. Line 354: Please provide further clarification, particularly if this information is intended to inform the development of future assays for *Aeromonas* species classification.

Done, please see line 400-408.

30. Line 364: Confirm whether a substantial number of ESBL genes were identified.

Done, please see lines 416-417.

31. Line 365: Replace with “AMR gene profile.”

Done.

32. Lines 365–374: Differences in AMR and virulence gene distribution may reflect the fact that all clinical isolates were from Pakistan, while environmental isolates were more diverse and came from two countries. This should be discussed.

Thank you for your comment. We have revised the discussion to make it clear that clinical isolates were all from Pakistan whereas the environmental isolates were from India and Bangladesh and noted that this sampling bias may have influenced the observed differences in gene distribution (see revised text, lines 441–443).

33. Line 387: Please provide more explanation on why isolates from Pakistan harbored a higher number of resistance genes. Could this be attributed to their clinical origin, where selection pressure from antibiotic use is typically higher? Alternatively, could the difference be influenced by the absence of environmental isolates from Pakistan in the study, limiting a direct comparison across sample types?

Added, please see lines 443-451.

34. Line 427: Please provide additional metadata for the isolates from Bangladesh.

Added, please see lines 492-494.

35. Line 501: This sentence is difficult to understand, please rewrite for clarity.

Thank you, we have rephrased this sentence (please see lines 574-575).

36. Line 807: Consider revising to: "Each dot is coloured based on the gene abundance per species (see scale), and the size of the dots corresponds to the number of genomes harboring the gene."

Done. Thank you

TITLE: *Aeromonas* in South Asia: Genomic Insights into an Environmental Pathogen and Reservoir of Antimicrobial Resistance.

Nature Communications manuscript NCOMMS-25-34237A

We are grateful to the reviewers for their thoughtful and constructive comments. Below, we provide detailed responses to each point raised and have revised the manuscript accordingly.

REVIEWER COMMENTS IN REVERSE ORDER FOR CLARITY

Reviewer #3 (Remarks to the Author):

I thank the authors for their thorough revision of the manuscript and for addressing my previous comments and questions. The data interpretation and overall clarity have been substantially improved. One minor comment remains: please revise the color scheme and legend of Supplementary Figure 5b to enhance clarity.

Thank you for your comment, we have revised Supplementary Figure 5b for improved clarity.

Reviewer #2 (Remarks to the Author):

The authors have resolved all my issues.

Many thanks for your suggestions on this manuscript.

Reviewer #1 (Remarks to the Author):

Dear Authors,

Thank you for the substantial and carefully executed revision of “*Aeromonas* in South Asia: Genomic Insights into an Environmental Pathogen and Reservoir of Antimicrobial Resistance.” I appreciate the scope of the study and the evident effort of the team, and I offer the following comments purely in a constructive spirit.

Thank you for your kind comments.

My remaining concern is the taxonomic identification of the *Aeromonas* isolates. Based on my research and experience with *Aeromonas*, average nucleotide identity (ANI) alone is not sufficient for species-level assignments, particularly in borderline cases. I strongly recommend that you combine ANI with digital DNA–DNA hybridization (dDDH), reporting both metrics and using accepted thresholds (e.g., ANI \approx 95–96% and dDDH \approx 70%) to reach stable conclusions. Please provide a table (preferably supplementary) listing, for each isolate, the closest type strain(s) together with ANI and dDDH values and the corresponding genome accessions, along with the software and parameters used (e.g., FastANI/pyani for ANI; GGDC/Formula 2 for dDDH).

As requested, the closest type strain(s) and corresponding ANI values for each isolate are included in Supplementary Data-2 as part of the GTDB-Tk output, which reports the nearest genome matches and associated ANI values for all isolates. For dDDH values please see below.

Relatedly, while GTDB/GTDB-Tk is valuable for genome-based taxonomy, it is not a nomenclatural authority and may include provisional or non-validated names and an incomplete set of type strains. In the current GTDB release you cite, I noted ~32 *Aeromonas* taxa irrespective of nomenclatural status, whereas the List of Prokaryotic names with Standing in Nomenclature (LPSN; <https://lpsn.dsmz.de/genus/aeromonas> [lpsn.dsmz.de]) currently recognises 34 validly published *Aeromonas* species. Please make sure your reference panel includes the type strain genomes of all valid species listed by LPSN, not only those represented in GTDB. Augmenting the GTDB-Tk reference with missing valid *Aeromonas* type strains may change some assignments, and this possibility should be checked and, if relevant, discussed.

Addressing these points will, I believe, make the taxonomic backbone of the manuscript more robust and increase confidence in the downstream ecological and AMR interpretations. Thank you again for your thorough work; I hope these suggestions are helpful.

We carefully considered the reviewer's suggestion regarding the inclusion of digital DNA–DNA hybridization (dDDH) values. As recommended, we used the Type (Strain) Genome Server (TYGS/ GGDC denoted TYGS from hence forth) linked from the LPSN web portal. Noting that TYGS allows only 50 genomes per submission and bootstrap-based confidence intervals are disabled when processing more than 20 genomes (see, GGDC FAQ here; hence not suitable for the 1,853 genomes analysed here), to address the reviewers comments directly we subsampled 50 phylogenetically representative genomes covering all species included in our study and assessed their dDDH values. Of the genomes analysed the species classification was identical for 42 of the 50 genomes analysed on either TYGS or GTDB-Tk (Table-1): indicating a high concordance between GGDC and GTDB-Tk classifications.

*Of the remaining eight genomes for 6 of these, the closest matching species (*A. veronii* & *A. allosaccharophila*) from TYGS was the same as the one determined by GTDB-Tk. However, due to borderline dDDH scores (*Aeromonas veronii* CECT 4257; $d_4 = 67.7\text{--}69.9$) they were either marked as unclassified or potential new species by TYGS (Table-1). Taking the borderline calls Hu et al. (2022, <https://doi.org/10.3389/fmicb.2022.910277>), suggest that when the dDDH value is close to, but below, the ~70% threshold, the taxonomic status of those strain should be confirmed using a combination of phenotypic, chemotaxonomic, and phylogenomic analyses. Given the two approaches above, and that all these genomes had an ANI>95% to the closest reference by GTDB-Tk it appears that the overall species identifications here were robust using the approaches described in our original analysis (consistent with Goris et al., 2007).*

*The 7th genome was only matched to the genus level by GTDB-Tk and assigned to 'a potential new species' by TYGS. The 8th genome was discordant between the two approaches, likely because the *A. rivipollensis* reference genome is not available in TYGS (For a summary of the dual taxa calls see attached Table-1).*

*It is also important to note that many of those with dDDH (formula d_4) values above 70% significantly matched more than one *Aeromonas* species, despite the deep phylogenetic divisions: For example, strain '23386_4#62', originally classified as *A. caviae*, shows a dDDH (d_4) value of 82.2% with *A. caviae* NCTC 12244 and 81.3% with *A. hydrophila* subsp. *anaerogenes* CECT 4221 (Table-1). Given the limitations of scalability for TYGS and that our approach, combining GTDB-Tk and FastANI, provides a standardised, reproducible, and computationally efficient framework that has been widely used for bacterial taxonomy, we respectfully argue not to include dDDH values into our analysis.*